

# Microscopic theory of fractional excitations in gapless quantum Hall states: semi-quantized quantum Hall states

**Oğuz Türker[1] and Tobias Meng [1]***

**1** Institut für Theoretische Physik and Würzburg-Dresden Cluster of Excellence ct.qmat,
Technische Universität Dresden, 01062 Dresden, Germany

* tobias.meng@tu-dresden.de

## Abstract

We derive the low-energy theory of semi-quantized quantum Hall states, a recently observed class of gapless bilayer fractional quantum Hall states. Our theory shows these states to feature gapless quasiparticles of fractional charge coupled to an emergent Chern-Simons gauge field. These gapless quasiparticles can be understood as composites of electrons and Laughlin-like quasiparticles. We show that semi-quantized quantum Hall states exhibit perfect interlayer drag, host non-Fermi liquid physics, and serve as versatile parent states for fully gapped topological phases hosting anyonic excitations.

# 1 Introduction

In recent years, topology has widely been recognized as an important organizing principle in nature. In its most simple form, topology allows to classify band structures of gapped, non-interacting systems [73, 74]. The role of an order parameter is played by so-called topological invariants: two systems can be smoothly deformed into one another if their topological invariants have identical values. In contrast, topologically distinct phases are separated by the closing and re-opening of a bulk gap.

This basic notion of topology has been extended in different ways. On the one hand, it has been recognized that gapless systems may also be topological. A prime example are Weyl semimetals, whose band structures have nodal points corresponding to quantized monopoles of Berry flux [72]. As a result, Weyl semimetals exhibit many of the characteristics of gapped topological phases, including topological edge states, and transport governed by quantum anomalies. On the other hand, the notion of topology has also been extended to strongly interacting systems. Of particular importance is the concept of topological order [20, 21], realized for example in fractional quantum Hall effects. Topologically ordered states have no analogue in non-interacting systems, exhibit long-range entanglement, fractional anyonic quasiparticles, and a topological ground state degeneracy on non-trivial manifolds.

While topological order requires a bulk gap, related phenomena have also been discussed in gapless systems such as the composite Fermi liquid in a half-filled Landau level [9, 55, 62], gapless quantum spin liquids [28, 29, 33, 38, 42–54, 56–59, 61, 63–67, 69], quantum Hall states related to non-unitary or nonrational conformal field theories [36, 37, 39–41, 70, 71], fractional Weyl semimetals [14, 19, 23, 31, 35], semimetals as parent of topologically ordered phases [16, 34], and others (see for example Refs. [15, 17, 32, 60]). A recent experiment has now identified a state that combines properties of a fractional quantum Hall effect with a gapless character [7]. This "semi-quantized quantum Hall state" has been measured in a bilayer quantum Hall system. Similar to a fractional quantum Hall state, it features quantized fractional Hall and Hall drag resistances alongside vanishing longitudinal resistances. In stark contrast to traditional bilayer quantum Hall states such as Halperin states [3, 21, 22], however, the semi-quantized quantum Hall state persists along a continuous line of filling factors. This compressible behaviour implies the absence of a full bulk gap.

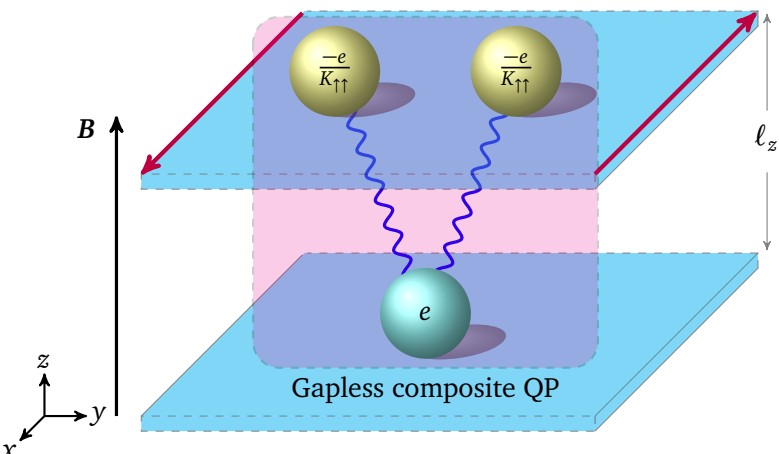

Figure 1: A semi-quantized quantum Hall state in a quantum Hall bilayer. The top layer is similar to a Laughlin state at filling factor $1/K_{\uparrow\uparrow}$, featuring a bulk gap and gapless edge states (indicated by red arrows). Gapless quasiparticles (QP) are composed of an electron in the bottom layer glued to Laughlin-like quasi-holes in the top layer by strong interlayer interactions. $\boldsymbol{B}$ is the magnetic field, $\ell_z$ the layer distance.

In this work, we derive and analyse the full low-energy theory of semi-quantized quantum Hall states using the formalism of coupled-wire constructions [18]. Our microscopic derivation of the topological field theory governing these states is an extension of the composite fermion picture of Ref. [7]. We go considerably beyond the theory put forward in that work, but agree with it where we overlap. Focussing on composite fermion filling factor one for simplicity, we find that semi-quantized quantum Hall states are composed of a gapped Laughlin-like sector coupled to fractionally charged, gapless quasiparticles via an emergent Chern-Simons gauge field, see Fig. 1. As a main advantage as compared to a continuum theory as the one in Sec. 2, our coupled-wire construction not only provides a microscopic model for semi-quantized quantum Hall states, but also facilitates an exact translation of observables between the languages of the effective low-energy field theory and the original electronic operators, which we use to analyse the intriguing properties of semi-quantized quantum Hall states. We find that transport features perfect interlayer drag, and in general exhibits non-Fermi liquid behaviour and a violated Wiedemann-Franz law. Our theory naturally explains the observed semi-quantized behavior, and shows that the Hall and Hall drag responses remain quantized on lines of fillings factors. This is remarkable since the system is composed of a gapped and a gapless layer. We discuss that the persistence of quantized responses arises only if the layer associated with the gapless sector is electrically disconnected. As one might expect, driving a current through that layer would instead lead to a non-quantized response. As an additional experimental signature, we show that semi-quantized quantum Hall states have a gapped electronic spectrum despite being globally gapless. Finally, fully gapped states deriving from semi-quantized quantum Hall states inherit their topologically non-trivial character. Consequently, even a simple charge-density wave gapped daughter state hosts anyonic excitations.

In a more general perspective, the present theory describes a gapless topological system combined with strong interactions, a class of problems for which many open questions remain. Proposing solvable Hamiltonians describing such phases, even in a controlled approximation, is a formidable challenge. The theory derived in this work provides a concrete, experimentally realizable scenario for a gapless system exhibiting topological-order like properties such as anyonic excitations, and fractional electro-magnetic responses. It details the description of such a phase in terms of a low-energy gauge theory, which quite naturally and transparently

emerges in the presently employed coupled-wire construction. Our theory paves the way for further analysis of gapless emergent gauge theories, for example in regard to the effects of gauge field disorder, and non-Fermi liquid physics.

The remainder of the paper is organized as follows. Sec. 2 provides a continuum version of the flux attachment scheme we use to describe semi-quantized quantum Hall states. This section serves to provide the reader with an intuition about the more exact mappings and transformations we later on perform in a coupled-wire language. In Sec. 3, we introduce a coupled-wire model for semi-quantized quantum Hall states, and transform the upper layer to a composite boson picture. This calculation closely mirrors the discussion of Refs. [5, 6] for single-layer systems. In Sec. 4, we develop the coupled-wire variant of inter-layer flux attachment for our system. Sec. 5 implements this transformation in the action, and derives the low-energy theory of semi-quantized quantum Hall states as the strong-coupling phase of sine-Gordon terms. This section is the technical backbone of our work. In Sec. 6, we study the properties of the gapless sector and show that the gapless quasiparticles can be understood as composites of electrons in the bottom layer and Laughlin quasi-holes in the top layer. In Sec. 7, we show that the action describing the gapless sector can be "refermionized" in a Luttinger liquid sense, and discuss the properties of this action, identical to the one obtained in Sec. 2 by a different approach. Sec. 8 discusses the electromagnetic response of semi-quantized quantum Hall states. Sec. 9 details new experimental fingerprints of semi-quantized quantum Hall states, including a gapped electronic spectrum, non-Fermi liquid physics, and fully gapped daughter states with anyonic quasiparticles. We finally conclude in Sec. 10.

## 2 Continuum theory

We begin with a heuristic continuum calculation outlining the flux attachment scheme that we more formally develop in the subsequent sections. This scheme is an extension of the composite fermion picture proposed in the study reporting the experimental observation of semi-quantized quantum Hall states [7], in which both layers were viewed in a composite fermion picture by attaching $(K_{\uparrow\uparrow} - 1)$ intra-layer fluxes to each electron in the top layer, and $K_{\uparrow\downarrow}$ interlayer fluxes to all electrons, with the flux quantum being $\Phi_0 = 2\pi/(-e)$, $K_{\uparrow\uparrow}$ being an odd integer, and $K_{\uparrow\uparrow}$ being any integer. Note that we call particles "bosons" or "fermions" with regard to their braiding statistics amongst themselves. As discussed in Ref. [7], the composite fermions in the top layer have the effective filing factors

$$\nu_{\uparrow,\text{eff}} = \frac{\nu_\uparrow}{1 - (K_{\uparrow\uparrow} - 1)\,\nu_\uparrow - \frac{q_\downarrow}{q_\uparrow} K_{\uparrow\downarrow}\,\nu_\downarrow}, \tag{1}$$

which for $\nu_{\uparrow,\text{eff}} = 1$ and $q_\uparrow = q_\downarrow = e$ agrees with the filling factor constraint that we find in our coupled wire approach, see Eq. (18) below. More generally, a semi-quantized quantum Hall state corresponds to an integer quantum Hall state of the composite fermions in one of the layers (the top layer in our model). The experimentally observed semi-quantized quantum Hall states for example live at filing factor $\nu_{\uparrow,\text{eff}} = 2$. As argued in Ref. [7], the Hall response of the composite fermions in the top layer is quantized if only the top layer is driven, reflecting the fact that the composite fermions in that layer form a gapped quantum Hall state. The constraint that the bottom layer does not carry a current leads to a quantized Hall drag resistance [7].

We now go beyond these arguments put forward in Ref. [7], and heuristically derive the full low-energy theory in a continuum flux attachment approach. It turns out that a composite boson picture is more appropriate for that purpose than a composite fermion picture. A quantum Hall state corresponds to a situation in which the effective magnetic field seen by the composite bosons is zero [25]. In that case, the system may form a superfluid coupled

to the external electromagnetic field and a statistical gauge field. The Higgs mechanism then "eats up" the gapless Goldstone mode and leads to an incompressible state (here, the sector described by the composite bosons will become incompressible) [26].

Our starting point are two isotropic and homogenous layers containing spin-polarized electrons. While the electrons are considered to interact strongly between each other (both within each layer, and between the layers), no tunnelling is allowed between the layers. In an external magnetic field in $z$-direction, the second-quantized Hamiltonian modelling this system reads [24]

$$
\begin{aligned}
H = &\sum_{\sigma=\uparrow,\downarrow} \int d^2\mathbf{x}\, \psi_\sigma^\dagger(\mathbf{x}) \left( \frac{(-i\partial_i + q_\sigma A_i)^2}{2m} + q_\sigma A_0 - \mu \right) \psi_\sigma(\mathbf{x}) \\
&+ \frac{1}{2} \int d^2\mathbf{x}\, \delta\hat\rho_\uparrow(\mathbf{x}) V(\mathbf{x}-\mathbf{y}) \delta\hat\rho_\uparrow(\mathbf{y}) + \int d^2\mathbf{x}\, \delta\hat\rho_\uparrow(\mathbf{x}) \widetilde{V}(\mathbf{x}-\mathbf{y}) \delta\hat\rho_\downarrow(\mathbf{y}),
\end{aligned}
\tag{2}
$$

where the index $\uparrow$ ($\downarrow$) labels electrons in top (bottom) layer, $A$ is the electromagnetic potential, $\mu$ is the chemical potential, $q_\sigma$ is charge of the electrons in layer $\sigma$, $\psi_\sigma$ is an electron annihilation operator in that layer, the repulsive intra (inter) layer interaction is $V$ ($\widetilde{V}$), and $\delta\hat\rho_\sigma = \hat\rho_\sigma - \rho_{\sigma,0}$ is the electronic density in layer $\sigma$ measured relative to a reference density $\rho_{\sigma,0}$. For simplicity, we assume the bottom layer electrons to not interact with each other (adding this interaction would not affect the qualitative argument put forward in this section). The index $i$ runs over the spatial coordinates $i = 1, 2$, and all vector fields are written in a covariant form using the summation notation.

At first, we apply a flux attachment transformation that maps the top layer electrons to composite hardcore bosons by attaching an odd number $K_{\uparrow\uparrow}$ of intra-layer fluxes in the top layer, and an integer number $K_{\uparrow\downarrow}$ of interlayer fluxes to both layers. The transformed Hamiltonian reads

$$
\begin{aligned}
H = &\int d^2\mathbf{x}\, \hat\phi^\dagger(\mathbf{x}) \left( \frac{(-i\partial_i + q_\uparrow A_i + \hat a_i^\uparrow[\hat\rho_\uparrow, \hat\rho_\downarrow])^2}{2m} + q_\uparrow A_0 - \mu \right) \hat\phi(\mathbf{x}) \Big] \\
&+ \int d^2\mathbf{x}\, \hat\psi^\dagger(\mathbf{x}) \left( \frac{(-i\partial_i + q_\downarrow A_i + \hat a_i^\downarrow[\hat\rho_\uparrow, \hat\rho_\downarrow])^2}{2m} + q_\downarrow A_0 - \mu \right) \hat\psi(\mathbf{x}) + H_{\text{int}},
\end{aligned}
\tag{3}
$$

where $\hat\phi$ denotes the composite boson annihilation operator in the top layer, while $\hat\psi$ is the composite fermion annihilation operator in the bottom layer. The functionals $\hat a_i^\sigma[\hat\rho_\uparrow, \hat\rho_\downarrow]$ satisfy

$$
\epsilon^{ij}\partial_i a_j^\uparrow[\hat\rho_\uparrow, \hat\rho_\downarrow] = -2\pi K_{\uparrow\uparrow}\hat\rho_\uparrow - 2\pi K_{\uparrow\downarrow}\hat\rho_\downarrow,
\tag{4a}
$$

$$
\epsilon^{ij}\partial_i a_j^\downarrow[\hat\rho_\uparrow, \hat\rho_\downarrow] = -2\pi K_{\uparrow\downarrow}\hat\rho_\uparrow.
\tag{4b}
$$

We then go to a functional integral picture, and promote $\hat a_i^\sigma$ to gauge fields by introducing Lagrangian multiplier fields $a_0^\sigma$ enforcing the constraints in Eqs. (4). The gauge-invariant

extension of the resulting theory reads

$$S = S_{\text{CB}} + S_{\text{CF}} + S_{\text{CS}} + S_{\text{hc}} + S_{\text{int}}, \tag{5a}$$

$$S_{\text{CB}} = \int d^{(2+1)}x \, \bar{\phi}\left(i\partial_0 - q_\uparrow A_0 - a_0^\uparrow - \frac{(-i\partial_i + q_\uparrow A_i + a_i^\uparrow)^2}{2m} + \mu\right)\phi, \tag{5b}$$

$$S_{\text{CF}} = \int d^{(2+1)}x \, \bar{\psi}\left(i\partial_0 - q_\downarrow A_0 - a_0^\downarrow - \frac{(-i\partial_i + q_\downarrow A_i + a_i^\downarrow)^2}{2m} + \mu\right)\psi, \tag{5c}$$

$$S_{\text{CS}} = -\frac{1}{4\pi K_{\uparrow\downarrow}}\int d^{(2+1)}x \, \epsilon^{\mu\nu\lambda} a_\mu^\downarrow \partial_\nu a_\lambda^\uparrow + \frac{K_{\uparrow\uparrow}}{4\pi K_{\uparrow\downarrow}^2}\int d^{(2+1)}x \, \epsilon^{\mu\nu\lambda} a_\mu^\downarrow \partial_\nu a_\lambda^\downarrow \tag{5d}$$

$$-\frac{1}{4\pi K_{\uparrow\downarrow}}\int d^{(2+1)}x \, \epsilon^{\mu\nu\lambda} a_\mu^\uparrow \partial_\nu a_\lambda^\downarrow, \tag{5e}$$

$$S_{\text{hc}} = -\int d^{(2+1)}x \left(\lambda_1 (\bar{\phi}\phi)^2 + \lambda_2 \, \bar{\psi}\psi \, \bar{\phi}\phi\right), \tag{5f}$$

where $S_{\text{hc}}$ describes the bosonic hard-core interaction, and $S_{\text{int}}$ contains the interactions $V$ and $\widetilde{V}$. Note that for clarity we have dropped the gauge fixing term, we used Landau gauge in our calculations.

Next, we perform an exact transformation for the composite bosons following Ref. [13],

$$\phi(x) = \sqrt{\rho_\uparrow(x)}e^{i\varphi(x)}\phi_{\text{v}}(x) \quad \text{and} \quad \bar{\phi}(x) = \sqrt{\rho_\uparrow(x)}e^{-i\varphi(x)}\phi_{\text{v}}^\dagger(x), \tag{6}$$

where the smooth part of the phase is denoted by $\varphi$, while $\phi_{\text{v}}(x)$ will describe vortex configurations of the phase (with $\phi_{\text{v}}^\dagger(x)\phi_{\text{v}}(x) = 1$). This transformation yields an efficient description of the low-energy physics if the amplitude $\sqrt{\rho_\uparrow}$ and the phases are physically relevant quantities, i.e. when the composite bosons in the top layer form a Bose condensate. In the usual treatment of quantum Hall states, one can infer the filling factors at which such a Bose condensate is stable from a mean-field analysis. In our case, this analysis is complicated by the fact that the composite bosons are still coupled to gapless fermionic degrees of freedom in the bottom layer. Motivated by the experimental observation that semi-quantized quantum Hall states feature an incompressible sub-sector, however, we from now on assume that we are in a situation where the composite bosons condense, and substitute Eq. (6) into Eq. (5). In this approximation, we heuristically assume that the vortices are point like. Neglecting spatial derivatives of $\rho_\uparrow$ based on the assumption that the composite bosons are condensed, the bosonic sector of the theory is described by an action $\int d^{2+1}x (\mathcal{L}_\phi + \mathcal{L}_{\phi,\text{int}})$ with

$$\mathcal{L}_\phi + \mathcal{L}_{\phi,\text{int}} = \rho_\uparrow(-\partial_0\varphi + i\phi_{\text{v}}^\dagger\partial_0\phi_{\text{v}} - q_\uparrow A_0 - a_0^\uparrow) - \rho_\uparrow\frac{(\partial_i\varphi - i\phi_{\text{v}}^\dagger\partial_i\phi_{\text{v}} + q_\uparrow A_i + a_i^\uparrow)^2}{2m}$$
$$-\frac{1}{2}\delta\rho_\uparrow V\delta\rho_\uparrow - \delta\rho_\uparrow \widetilde{V}\delta\rho_\downarrow + \mu\rho_\uparrow - \lambda_1\rho_\uparrow^2 - \lambda_2\rho^\uparrow\rho^\downarrow. \tag{7}$$

We then perform the Hubbard-Stratonovich transformation [13]

$$-\rho_\uparrow\frac{(\partial_i\varphi - i\phi_{\text{v}}^\dagger\partial_i\phi_{\text{v}} + q_\uparrow A_i + a_i^\uparrow)^2}{2m} \rightarrow -\frac{m}{2\rho_\uparrow^0}J^i J_i - J^i(\partial_i\varphi - i\phi_{\text{v}}^\dagger\partial_i\phi_{\text{v}} + q_\uparrow A_i + a_i^\uparrow), \tag{8}$$

where $J_i$ is a bosonic Hubbard-Stratonovich field. Integrating out the field $\varphi$ shows that the Hubbard-Stratonovich field satisfies the continuity equation $\partial_\mu J^\mu = 0$, where we defined $J^0 = \rho_\uparrow$. We can satisfy this constraint by introducing yet another gauge field $\alpha$ that relates

to $J$ as $J^\mu = \epsilon^{\mu\nu\lambda}\partial_\nu \alpha_\lambda/2\pi$. We plug this definition into the action, and neglect terms that contain more than one derivative of $\alpha$. This yields

$$\mathcal{L}_\phi + \mathcal{L}_{\phi,\text{int}} = -\alpha_\mu j^\mu_{\text{qp}} + \frac{1}{2\pi}\epsilon^{\mu\nu\lambda}\alpha_\mu\partial_\nu(-q_\uparrow A_\lambda - \alpha^\uparrow_\lambda) - \delta\rho_\uparrow \widetilde{V}\delta\rho_\downarrow + \mu\rho_\uparrow - \lambda_2 \rho^\uparrow \rho^\downarrow, \quad (9)$$

where we defined the 3-current $j^\mu_{\text{qp}} = \frac{1}{2\pi i}\epsilon^{\mu\nu\lambda}\partial_\nu(\phi^\dagger_v\partial_\lambda\phi_v)$, which physically corresponds to the current of quasiparticles in the gapped sector [13]. The emergent statistical gauge field $a^\uparrow_\mu$ enters the total action only linearly. When we integrate it out, we obtain the constraint

$$\epsilon^{\mu\nu\lambda}\partial_\nu\alpha_\lambda = -\frac{1}{K_{\uparrow\downarrow}}\epsilon^{\mu\nu\lambda}\partial_\nu a^\downarrow_\mu. \quad (10)$$

Next, we also integrate over the field $a^\downarrow_\mu$. By virtue of the constraint in Eq. (10), this leads to $\mathcal{S} = \int dx\, \mathcal{L}$ with

$$\begin{aligned}
\mathcal{L} = &-\alpha_\mu j^\mu_{\text{qp}} - \frac{q_\uparrow}{2\pi}\epsilon^{\mu\nu\lambda}A_\mu\partial_\nu\alpha_\lambda + \frac{K_{\uparrow\uparrow}}{4\pi}\epsilon^{\mu\nu\lambda}\alpha_\mu\partial_\nu\alpha_\lambda \\
&+ \bar\psi\left(i\partial_0 - q_\downarrow A_0 + K_{\uparrow\downarrow}\,\alpha_0 - \frac{(-i\partial_i + q_\downarrow A_i - K_{\uparrow\downarrow}\,\alpha_i)^2}{2m} + \mu\right)\psi \\
&+ \mathcal{L}_{\text{ints}},
\end{aligned} \quad (11)$$

where $\mathcal{L}_{\text{ints}}$ comprises all remaining interaction terms. Upon shifting the emergent gauge field as $\beta_\mu = \alpha_\mu - A_\mu(q_\uparrow/K_{\uparrow\uparrow})$, we obtain the action

$$\begin{aligned}
\mathcal{L} = &-\beta_\mu j^\mu_{\text{qp}} + \frac{K_{\uparrow\uparrow}}{4\pi}\epsilon^{\mu\nu\lambda}\beta_\mu\partial_\nu\beta_\lambda - \frac{q_\uparrow}{K_{\uparrow\uparrow}}j^\mu_{\text{qp}}A_\mu - \frac{q^2_\uparrow}{4\pi K_{\uparrow\uparrow}}\epsilon^{\mu\nu\lambda}A_\mu\partial_\nu A_\lambda \\
&+ \bar\psi\left(i\partial_0 - q^*_\downarrow A_0 + K_{\uparrow\downarrow}\beta_0 - \frac{(-i\partial_i + q^*_\downarrow A_i - K_{\uparrow\downarrow}\beta_i)^2}{2m} + \mu\right)\psi + \mathcal{L}_{\text{ints}},
\end{aligned} \quad (12)$$

where $q^*_\downarrow = q_\downarrow - q_\uparrow K_{\uparrow\downarrow}/K_{\uparrow\uparrow}$. Eq. (12) is the full low-energy theory describing semi-quantized quantum Hall states: a topological field theory of a gapless sector containing fractionally charged quasiparticles coupled to an emergent Chern-Simons gauge field. The coupling to the emergent gauge field implies that the fractionally charged quasiparticles form an anyonic liquid. In the following sections, we re-derive Eq. (12) more formally using a coupled-wire construction, before then analyzing its properties.

## 3 Microscopic coupled-wire model and composite boson picture in the top layer

Our coupled-wire model starts by deforming the bilayer of two-dimensional electrons measured in the experiment [7] into a bilayer of quantum wires, see Fig. 2. This transformation leaves topological properties of Hall effects invariant [18]. Neighbouring wires within a layer, spaced by $\ell_y$, are weakly tunnel-coupled, but there is no tunnelling between the layers. Each wire hosts a single electronic band. Using Landau gauge, the dispersion of the $j$-th wire in layer $\sigma = \uparrow, \downarrow$ is

$$\mathcal{E}_{j\sigma}(k_x) = (k_x + bj)^2/2m, \quad (13)$$

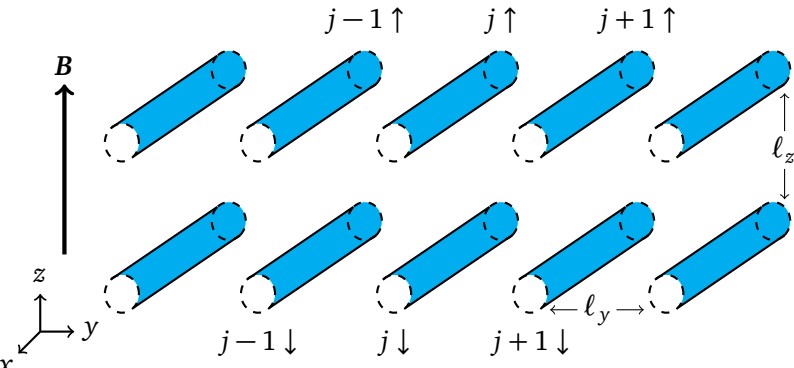

Figure 2: The coupled-wire bilayer model. Each wire is labelled by a wire number $j$ and a layer index $\sigma = \uparrow, \downarrow$, the distance between adjacent wires within a layer is $\ell_y$.

with $b = eB\ell_y$. Here, $m$ is the effective mass, $e < 0$ denotes the electron charge, and $B > 0$ is the magnetic field [1]. At low energies, only the right-moving and left-moving modes close to the Fermi level are important. The electronic operators can then be decomposed as

$$\psi_{j\sigma} \approx e^{ik_{F,Rj\sigma}}\psi_{Rj\sigma} + e^{ik_{F,Lj\sigma}}\psi_{Lj\sigma}, \tag{14}$$

where $\psi_{rj\sigma}$ annihilates a right ($r = R$) or left ($r = L$) mover. Considering identical filling of all wires within a layer, the Fermi momenta can be written as $k_{F,rj\sigma} = b\,j + \hat{r}\,k_{F\sigma}^0$ with $\hat{R} = +1$ and $\hat{L} = -1$. The electronic density per layer is $\rho_\sigma = k_{F\sigma}^0/\pi\ell_y$, which translates to filling factors $\nu_\sigma = 2k_{F\sigma}^0/b$. The non-interacting low-energy physics can thus be modelled by the Hamiltonian

$$H_0 = -iv_{\mathrm{F}} \int dx \sum_{\sigma,j} \left[ \psi_{Rj\sigma}^\dagger \partial_x \psi_{Rj\sigma} - \psi_{Lj\sigma}^\dagger \partial_x \psi_{Lj\sigma} \right], \tag{15}$$

where $v_F$ denotes the Fermi velocity. Following Ref. [4], we bosonize the chiral modes as $\psi_{rj\sigma} = (U_{rj\sigma}/\sqrt{2\pi\alpha})\exp\{-i\Phi_{rj\sigma}\}$. Here, $\alpha^{-1}$ is a large momentum cut-off, while $U_{rj\sigma}$ is a Klein factor that can safely be ignored in the remainder [1, 2]. The bosonized fields obey

$$[\Phi_{rj\sigma}(x), \Phi_{r'j'\sigma'}(x')] = \delta_{rr'}\delta_{jj'}\delta_{\sigma\sigma'}\,i\pi\hat{r}\,\mathrm{sgn}(x - x'), \tag{16}$$

and relate to density fluctuations as $\rho_{rj\sigma} - \langle\rho_{rj\sigma}\rangle = -\frac{\hat{r}}{2\pi}\partial_x\Phi_{rj\sigma}$. The chiral fields can alternatively be represented as $\Phi_{rj\sigma} = \hat{r}\phi_{j\sigma} - \theta_{j\sigma}$. After bosonization, the non-interacting Hamiltonian becomes

$$H_0 = \frac{v_F}{2\pi} \int dx \sum_{j,\sigma} \left[ (\partial_x\phi_{j\sigma})^2 + (\partial_x\theta_{j\sigma})^2 \right]. \tag{17}$$

Electron-electron interaction modify this Hamiltonian in two ways: scatterings that do not transfer particles between chiral channels (forward scatterings) change $v_F$ to effective velocities. Other interactions (backscatterings) open gaps.

As was suggested experimentally, semi-quantized quantum Hall states form by a partial gap closing in Halperin bilayer states [7]. A Halperin bilayer state forms if composite fermions in both the top and the bottom layer form fully gapped quantum Hall states. In the language of coupled-wire constructions [3], these bilayer composite fermion quantum Hall states correspond to the strong-coupling phase of two families of correlated inter-wire backscatterings (one per layer). These backscatterings can only be relevant in the renormalization group (RG)

---

[1]We use units such that $\hbar = 1$ and $c = 1$.

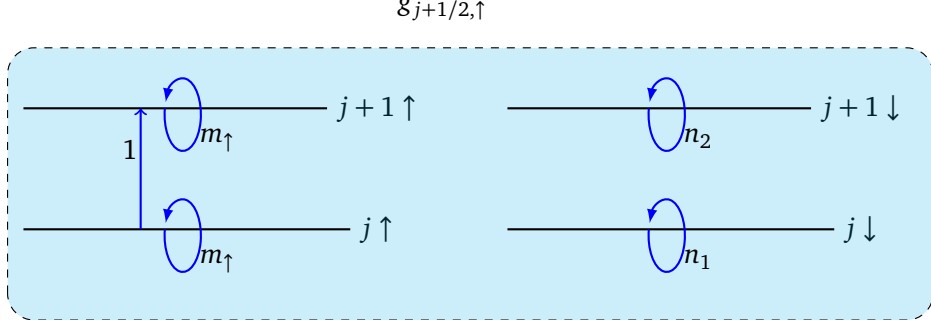

Figure 3: Correlated inter-wire backscattering stabilizing a semi-quantized quantum Hall state. Arrows indicate tunnelling between and backscattering within wires. The numbers of scattered electrons is indicated besides each arrow.

sense if they conserve momentum. Each family of backscatterings thus imposes one condition on the fillings, which in turn completely determines the filling factors $(\nu_\uparrow^0, \nu_\downarrow^0)$ of a given Halperin bilayer state. If the filling factors are detuned from $(\nu_\uparrow^0, \nu_\downarrow^0)$, the system can enter a semi-quantized quantum Hall states for combinations of filling factors at which one, but not both families of backscatterings preserve momentum.

In the remainder, we consider the correlated inter-wire backscatterings in the top layer to preserve momentum. As illustrated in Fig. 3, this process corresponds to an electron tunnelling between wires $j$ and $j+1$ in the top layer, while simultaneous backscatterings in wires $j$ and $j+1$ of both layers ensure momentum conservation. More precisely, momentum is conserved if [3]

$$K_{\uparrow\uparrow}\, \nu_\uparrow + K_{\uparrow\downarrow}\, \nu_\downarrow = 1, \tag{18}$$

where $K_{\uparrow\uparrow} = 1 + 2m_\uparrow$ and $K_{\uparrow\downarrow} = n_1 + n_2$. In a bosonized language, these backscatterings correspond to the sine-Gordon Hamiltonian

$$H_{g\uparrow} = \sum_j \int dx\, g_{j+1/2\uparrow} \cos\left(\widetilde{\Phi}_{Rj\uparrow} - \widetilde{\Phi}_{Lj+1\uparrow}\right), \tag{19}$$

where we have introduced the fields

$$\widetilde{\Phi}_{Rj\sigma} = (1+m_\sigma)\Phi_{Rj\sigma} - m_\sigma \Phi_{Lj\sigma} + n_1(\Phi_{Rj\bar\sigma} - \Phi_{Lj\bar\sigma}), \tag{20a}$$

$$\widetilde{\Phi}_{Lk\sigma} = (1+m_\sigma)\Phi_{Lj\sigma} - m_\sigma \Phi_{Rj\sigma} - n_2(\Phi_{Rj\bar\sigma} - \Phi_{Lj\bar\sigma}), \tag{20b}$$

with $\bar\uparrow = \downarrow$ and vice versa. The values of $m_\uparrow$, $n_1$ and $n_2$ are set by the backscatterings, whereas $m_\downarrow$ is a free parameter that drops out at the end of the calculation. These fields obey

$$[\widetilde{\Phi}_{rj\sigma}(x), \widetilde{\Phi}_{r'j'\sigma'}(x')] = \delta_{rr'}\delta_{jj'} K_{\sigma\sigma'}\, i\pi\hat{r}\, \text{sgn}(x-x'), \tag{21}$$

where the $K$-matrix reads

$$K = \begin{pmatrix} K_{\uparrow\uparrow} & K_{\uparrow\downarrow} \\ K_{\uparrow\downarrow} & K_{\downarrow\downarrow} \end{pmatrix} = \begin{pmatrix} 1+2m_\uparrow & n_1+n_2 \\ n_1+n_2 & 1+2m_\downarrow \end{pmatrix}. \tag{22}$$

Note that the $K$-matrix satisfies $K^T = K$.

We now switch to a more convenient covariant notation [2], and keep track of small deviations $\mathcal{A}$ of the electromagnetic potentials $A$ from $A_1 = -eBy$ and $A_0 = A_2 = 0$ in the gauge $\mathcal{A}_1 = 0$. Writing the unit of charge in layer $\sigma$ as $q_\sigma$, with physical values $q_\uparrow = q_\downarrow = e$, yields the action

$$S = \sum_{j,\sigma} \int d^{1+1}x \left[ \frac{1}{\pi}(\partial_0 \theta_{j\sigma} + q_\sigma \mathcal{A}_{0,j})(\partial_x \phi_{j\sigma}) - \frac{u_\sigma}{2\pi}(\partial_x \phi_{j\sigma})^2 - \frac{v_\sigma}{2\pi}(\partial_x \theta_{j\sigma})^2 \right.$$
$$\left. - g_{j+1/2\uparrow} \cos\left( \widetilde{\Phi}_{Rj\sigma} - \widetilde{\Phi}_{Lj+1\sigma} + q_\uparrow \ell_y \mathcal{A}_{2,j+1/2} \right) \right], \tag{23}$$

where $\ell_y \mathcal{A}_{2,j+1/2} = \int_{j \to j+1} dy' \, \mathcal{A}_2$, while $u_\sigma$ and $v_\sigma$ denote effective Luttinger liquid velocities (both are of the order of $v_F$).

## 3.1 Composite boson picture for the ↑-degrees of freedom

To derive the low-energy theory in the strong-coupling phase of $H_{g\uparrow}$, we now perform a sequence of exact transformations. We begin by transforming the top layer to a composite boson picture. To that end, we first rewrite the action in a "diagonal basis" by defining operators that commute between the gapped and gapless sectors. The "diagonal basis" is given by the fields $\widetilde{\Phi}_{rj\uparrow}(x)$ and $\Theta_{rj\downarrow}(x)$ defined as

$$\Theta_{rj\downarrow} = K_{\downarrow\uparrow}^{-1} \widetilde{\Phi}_{rj\uparrow} + K_{\downarrow\downarrow}^{-1} \widetilde{\Phi}_{rj\downarrow} \,. \tag{24}$$

By construction, these fields obey

$$[\Theta_{rj\downarrow}(x), \Theta_{r'k'\downarrow}(x')] = \delta_{rr'} \delta_{kk'} K_{\downarrow\downarrow}^{-1} i\pi \hat{r} \operatorname{sgn}(x - x') \quad \text{and} \quad [\widetilde{\Phi}_{rj\uparrow}(x), \Theta_{r'j'\downarrow}(x')] = 0. \tag{25}$$

Operators containing only the fields $\Theta_{rj\downarrow}$ do thus not affect the ↑-sector. Next, we perform a second transformation of the fields by defining

$$\widehat{\phi}_{j\uparrow}(x) = \frac{\widetilde{\Phi}_{Rj\uparrow}(x) - \widetilde{\Phi}_{Lj\uparrow}(x)}{2}, \tag{26a}$$

$$\widehat{\theta}_{j\uparrow}(x) = \frac{-\widetilde{\Phi}_{Rj\uparrow}(x) - \widetilde{\Phi}_{Lj\uparrow}(x)}{2}, \tag{26b}$$

$$\widehat{\phi}_{j\downarrow}(x) = \frac{\Theta_{Rj\downarrow}(x) - \Theta_{Lj\downarrow}(x)}{2}, \tag{26c}$$

$$\widehat{\theta}_{j\downarrow}(x) = \frac{-\Theta_{Rj\downarrow}(x) - \Theta_{Lj\downarrow}(x)}{2}. \tag{26d}$$

Upon choosing the forward scattering terms in $\mathcal{S}_{\text{fs}}$ appropriately, the action then takes the form

$$\mathcal{S} = \sum_{\sigma=\uparrow,\downarrow} \mathcal{S}_{2,\sigma} + \mathcal{S}_{\text{fs}} + \mathcal{S}_{\text{SG}} = \widehat{\mathcal{S}}_{2,\sigma} + \mathcal{S}_{\text{SG}}, \tag{27a}$$

$$\widehat{\mathcal{S}}_{2,\uparrow} = \int d^{1+1}x \sum_j \left[ \frac{1}{\pi K_{\uparrow\uparrow}}(\partial_t \widehat{\theta}_{j\uparrow} + q_\uparrow \mathcal{A}_{0,j})(\partial_x \widehat{\phi}_{j\uparrow}) - \frac{\widehat{u}_\uparrow}{2\pi}(\partial_x \widehat{\phi}_{j\uparrow})^2 - \frac{\widehat{v}_\uparrow}{2\pi}(\partial_x \widehat{\theta}_{j\uparrow})^2 \right], \tag{27b}$$

$$\mathcal{S}_{\text{SG}} = \int d^{1+1}x \sum_j (-g_{\uparrow,j+1/2}) \cos\left( \widetilde{\Phi}_{Rj\uparrow} - \widetilde{\Phi}_{Lj+1\uparrow} - q_\uparrow \ell_y \mathcal{A}_{y,j+1/2} \right). \tag{27c}$$

---

[2]We choose the metric such that time is $t = x^0 = x_0$, while the spatial coordinates are $x = x^1 = -x_1$, and $y = x^2 = -x_2$.

Next, we define composite boson fields in the $\uparrow$-layer. Following Refs. [5, 6], this is done by introducing

$$\phi_j^{\mathrm{CB}}(x) = \frac{1}{K_{\uparrow\uparrow}} \frac{\widetilde{\Phi}_{Rj\uparrow}(x) - \widetilde{\Phi}_{Lj\uparrow}(x)}{2} = \frac{1}{K_{\uparrow\uparrow}} \widehat{\phi}_{j\uparrow}(x), \tag{28a}$$

$$\theta_j^{\mathrm{CB}}(x) = -\frac{\widetilde{\Phi}_{Rj\uparrow}(x) + \widetilde{\Phi}_{Lj\uparrow}(x)}{2} - \sum_{j'\neq j} \mathrm{sgn}(j'-j) \frac{\widetilde{\Phi}_{Rj'\uparrow}(x) - \widetilde{\Phi}_{Lj'\uparrow}(x)}{2}, \tag{28b}$$

$$= \widehat{\theta}_{j\uparrow} - K_{\uparrow\uparrow} \sum_{j'\neq j} \mathrm{sgn}(j'-j) \phi_{j'}^{\mathrm{CB}}. \tag{28c}$$

The action can now be written as

$$\mathcal{S} = \mathcal{S}_{2,\mathrm{CB}} + \widehat{\mathcal{S}}_{2,\downarrow} + \mathcal{S}_{\mathrm{SG}}, \tag{29a}$$

$$\mathcal{S}_{2,\mathrm{CB}} = \int d^{1+1}x \sum_j \Bigg[ \frac{1}{\pi} (\partial_t \theta_k^{\mathrm{CB}} + q_\uparrow \mathcal{A}_{0,j})(\partial_x \phi_k^{\mathrm{CB}})$$
$$- \frac{u_{\mathrm{CB}}}{2\pi} (\partial_x \phi_k^{\mathrm{CB}})^2 - \frac{v_{\mathrm{CB}}}{2\pi} (\partial_x \theta_k^{\mathrm{CB}} + a_k)^2 \Bigg], \tag{29b}$$

$$\widehat{\mathcal{S}}_{2,\downarrow} = \int d^{1+1}x \sum_j \Bigg[ \frac{1}{\pi K_{\downarrow\downarrow}^{-1}} (\partial_t \widehat{\theta}_{j\downarrow} + K_{\downarrow\downarrow}^{-1} q_\downarrow^* \mathcal{A}_{0,j})(\partial_x \widehat{\phi}_{j\downarrow})$$
$$- \frac{\widehat{u}_\downarrow}{2\pi} (\partial_x \widehat{\phi}_{j\downarrow})^2 - \frac{\widehat{v}_\downarrow}{2\pi} (\partial_x \widehat{\theta}_{j\downarrow})^2 \Bigg], \tag{29c}$$

$$\mathcal{S}_{\mathrm{SG}} = \int d^{1+1}x \sum_j (-g_{\uparrow,j+1/2}) \cos\Big( \theta_{k+1}^{\mathrm{CB}} - \theta_k^{\mathrm{CB}} - q_\uparrow \ell_y \mathcal{A}_{y,j+1/2} \Big), \tag{29d}$$

where we have introduced the abbreviation

$$a_j = \sum_{j'\neq j} \mathrm{sgn}(j'-j) \partial_x \widehat{\phi}_{j'\uparrow}(x) = K_{\uparrow\uparrow} \sum_{j'\neq j} \mathrm{sgn}(j'-j) \partial_x \phi_{j'}^{\mathrm{CB}}(x), \tag{30}$$

as well as $u_{\mathrm{CB}} = K_{\uparrow\uparrow}^2 \widehat{u}_\uparrow$, $v_{\mathrm{CB}} = \widehat{v}_\uparrow$.

## 3.2 Vortex duality transformation

Our next step is to perform the vortex duality transformation in the $\uparrow$-sector by defining [5, 6]

$$\phi_{j+1/2}^{\mathrm{VCB}}(x) = \frac{\theta_{j+1}^{\mathrm{CB}}(x) - \theta_j^{\mathrm{CB}}(x)}{2}, \tag{31}$$

$$\theta_{j+1/2}^{\mathrm{VCB}}(x) = \sum_{j'} \mathrm{sgn}(j'-j-1/2) \phi_{j'}^{\mathrm{CB}}(x). \tag{32}$$

We furthermore introduce

$$\widetilde{\alpha}_j = -\frac{1}{K_{\uparrow\uparrow}} \sum_{j'} \mathrm{sgn}(j'-j+1/2) \partial_x \phi_{j'+1/2}^{\mathrm{VCB}}, \tag{33}$$

which implies $\widetilde{\alpha}_j = \frac{1}{K_{\uparrow\uparrow}} \partial_x \theta_j^{\mathrm{CB}}(x)$ and $\widetilde{\alpha}_{j+1}(x) - \widetilde{\alpha}_j(x) = \frac{2}{K_{\uparrow\uparrow}} \partial_x \phi_{j+1/2}^{\mathrm{VCB}}(x)$. After a bit of algebra, and introducing

$$S\widetilde{\alpha}_{j+1/2} = \frac{\widetilde{\alpha}_{j+1} + \widetilde{\alpha}_j}{2} \quad \text{and} \quad \Delta\theta_j^{\mathrm{VCB}} = \theta_{j+1/2}^{\mathrm{VCB}} - \theta_{j-1/2}^{\mathrm{VCB}}, \tag{34}$$

the vortex duality implies the replacement $\mathcal{S}_{2,\text{CB}} \to \mathcal{S}_{2,\text{VCB}}$ with

$$
\begin{aligned}
\mathcal{S}_{2,\text{VCB}} = \int d^{1+1}x \sum_j \Bigg[ &\frac{1}{\pi}\left(\partial_t \theta^{\text{VCB}}_{j+1/2}\right)\left(\partial_x \phi^{\text{VCB}}_{j+1/2}\right) + \frac{1}{\pi}q_\uparrow \frac{\Delta\mathcal{A}_{0,j+1/2}}{2}\left(\partial_x \theta^{\text{VCB}}_{j+1/2}\right) \\
&- \frac{v_{\text{CB}}}{2\pi}\left(\partial_x \phi^{\text{VCB}}_{j+1/2}(x)\right)^2 - \frac{v_{\text{CB}}}{2\pi}\left(\partial_x \phi^{\text{VCB}}_{j+1/2}(x)\right)^2 - \frac{u_{\text{CB}}-K_{\uparrow\uparrow}^2 v_{\text{CB}}}{8\pi}(\partial_x \Delta\theta^{\text{VCB}}_j)^2 \\
&- \frac{v_{\text{CB}}K_{\uparrow\uparrow}^2}{2\pi}\left(\partial_x \theta^{\text{VCB}}_{j+1/2} + S\widetilde{\alpha}_{j+1/2}\right)^2 \Bigg].
\end{aligned}
\tag{35}
$$

# 4 Inter-layer flux attachment and local form of quasiparticle hopping

So far, the bottom layer is still described by the fields $\Theta_{rj\downarrow}$ introduced in Eq. (24). Because these fields commute with the fields in the $\uparrow$-sector, one might be tempted to define quasiparticle operators in the gapless $\downarrow$-sector as

$$
\chi_{rj\downarrow} \overset{?}{=} \exp\{i\Theta_{rj\downarrow}\}.
\tag{36}
$$

Hopping between neighbouring wires would then be represented by the operator

$$
\mathcal{T}_{r,r',j,\downarrow} = \chi^\dagger_{rj+1\downarrow}\chi_{r'j\downarrow},
\tag{37}
$$

which has a very non-local form. By that, we mean that they cannot be represented by a product of a small number of original electronic operators within a restricted region of space, since for example

$$
\Theta_{Rj\downarrow} - \Theta_{Lj+1\downarrow} = K_{\downarrow\downarrow}^{-1}\left(\widetilde{\Phi}_{Rj\downarrow} - \widetilde{\Phi}_{Lj+1\downarrow}\right) + K_{\downarrow\uparrow}^{-1}\left(\widetilde{\Phi}_{Rj\uparrow} - \widetilde{\Phi}_{Lj+1\uparrow}\right)
$$

contains the non-integer numbers $K_{\sigma\sigma'}^{-1}$. This is unsatisfactory for two reasons: first, the free parameter $m_\downarrow$ entering $K_{\sigma\sigma'}^{-1}$ seems to be relevant to this discussion. Second, we would like hopping of quasiparticles to be represented by a local operator in the above sense in order for it to be generated at a reasonable order in perturbation theory in the electronic tunnelling in the bottom layer and electron-electron interactions. In addition, we are still missing the interlayer flux attachment. As we show now, the latter resolves the former issues. We find that the correct transformation that attaches interlayer fluxes to both layers, preserves the fact that operators in the two sectors commute with one another, and provides a local expression for quasiparticle hopping is given by

$$
\widetilde{\phi}^{\text{VCB}}_{j+1/2}(x) = \phi^{\text{VCB}}_{j+1/2}(x),
\tag{38a}
$$

$$
\widetilde{\theta}^{\text{VCB}}_{j+1/2}(x) = \theta^{\text{VCB}}_{j+1/2}(x) - \frac{K_{\uparrow\downarrow}}{K_{\uparrow\uparrow}}\sum_{j'}\text{sgn}(j'-j-1/2)\,\widehat{\phi}_{j'\downarrow}(x)
\tag{38b}
$$

$$
= \theta^{\text{VCB}}_{j+1/2}(x) + \frac{K_{\downarrow\uparrow}^{-1}}{K_{\downarrow\downarrow}^{-1}}\sum_{j'}\text{sgn}(j'-j-1/2)\,\widehat{\phi}_{j'\downarrow}(x),
$$

$$
\widetilde{\phi}_{j\downarrow}(x) = \widehat{\phi}_{j\downarrow}(x),
\tag{38c}
$$

$$
\begin{aligned}
\widetilde{\theta}_{j\downarrow}(x) = \frac{1}{K_{\downarrow\downarrow}^{-1}}\Bigg( &\widehat{\theta}_{j\downarrow}(x) + K_{\downarrow\uparrow}^{-1}\sum_{j'}\text{sgn}(j'-j+1/2)\,\phi^{\text{VCB}}_{j'+1/2}(x) \\
&- K_{\downarrow\uparrow}^{-1}K_{\uparrow\downarrow}\sum_{j'\neq j}\text{sgn}(j'-j)\,\widehat{\phi}_{j'\downarrow}(x)\Bigg),
\end{aligned}
\tag{38d}
$$

where we used $K_{\uparrow\downarrow}^{-1}/K_{\downarrow\downarrow}^{-1} = -K_{\uparrow\downarrow}/K_{\uparrow\uparrow}$. These fields obey the canonical commutators

$$[\widetilde{\phi}_{j+1/2}^{\mathrm{VCB}}(x), \widetilde{\theta}_{j'+1/2}^{\mathrm{VCB}}(x')] = \delta_{jj'}\frac{i\pi}{2}\,\mathrm{sgn}(x'-x) \tag{39a}$$

$$[\widetilde{\phi}_{j\downarrow}(x), \widetilde{\theta}_{j'\downarrow}(x')] = \delta_{jj'}\frac{i\pi}{2}\,\mathrm{sgn}(x'-x), \tag{39b}$$

while all other commutators vanish.

## 4.1 Backscattering within wires, and tunnelling between wires

We now illustrate that the transformations in Eq. (38) indeed lead to local expressions for quasiparticle tunnelling. We define quasiparticles in the gapless ↓-sector as being created by exponentials of the chiral fields $\widetilde{\Theta}'_{rj\downarrow} = r\,\widetilde{\phi}_{j\downarrow} - \widetilde{\theta}_{j\downarrow}$. While the creation of an individual quasiparticle is highly non-local in terms of the original fermions, all string factors cancel for backscattering and hopping between neighbouring wires, and the remaining terms can be written as integer combination of the original electronic fields.

### 4.1.1 Backscattering of quasiparticles within a wire

Backscattering of quasiparticles within wire $j$ is implemented by $\exp\left\{\widetilde{\Theta}'_{Rj\downarrow} - \widetilde{\Theta}'_{Lj\downarrow}\right\}$. To see that this operator takes a local form in the above sense, we use Eqs. (38d), (38c), (26c), (24) and 20 to obtain

$$\widetilde{\Theta}'_{Rj\downarrow} - \widetilde{\Theta}'_{Lj\downarrow} = \Phi_{Rj\downarrow} - \Phi_{Lj\downarrow}. \tag{40}$$

Backscattering of quasiparticles in the gapless ↓-sector is thus the same as backscattering of the original electrons.

### 4.1.2 Tunnelling of quasiparticles between neighbouring wires

Tunnelling of quasiparticles between wires $j$ and $j+1$ is in general implemented by $\exp\left\{\widetilde{\Theta}'_{rj+1\downarrow} - \widetilde{\Theta}'_{r'j\downarrow}\right\}$. As for the backscattering, we can rewrite this operator using the definitions of the various fields. We also recall that $K^{-1}K = \mathbb{1}$ implies $K_{\downarrow\uparrow}^{-1}K_{\uparrow\downarrow} + K_{\downarrow\downarrow}^{-1}K_{\downarrow\downarrow} = 1$. With some algebra, we obtain

$$\widetilde{\Theta}'_{rj+1\downarrow} - \widetilde{\Theta}'_{r'j\downarrow} = \hat{r}\,\widetilde{\phi}_{k+1\downarrow} - \widetilde{\theta}_{k+1\downarrow} - \hat{r}'\,\widetilde{\phi}_{j\downarrow} + \widetilde{\theta}_{j\downarrow}$$

$$= \Phi_{rj+1\downarrow} - \Phi_{r'j\downarrow} - n_2(\Phi_{Rj\uparrow} - \Phi_{Lj\uparrow}) - n_1(\Phi_{Rj+1\uparrow} - \Phi_{Lj+1\uparrow}). \tag{41}$$

We thus find that the quasiparticles defined by exponentials of the new chiral fields $\widetilde{\Theta}'_{rj\downarrow}$ are tunnelled by local operators involving the hopping of an electron in the ↓-layer and the backscattering of electrons in the ↑-layer. Since it is known that tunnelling of Laughlin-like quasiparticles in the top layer is implemented by backscatterings of the original electrons there [2], this fits a picture of the gapless quasiparticles as composites of electrons in the bottom layer and Laughlin-like quasiparticles in the top layer.

# 5 Low-energy effective theory

To find the low-energy action of semi-quantized quantum Hall states, we plug to the fields introduced in Eqs. (38a) through (38d) into the action. In doing so, it is convenient to introduce a short-hand notation for the string factor appearing in the definition of $\tilde{\theta}_{j\downarrow}$. We thus define

$$\alpha_j(x) = -\frac{1}{K_{\uparrow\uparrow}} \sum_{j'} \text{sgn}(j'-j+1/2)\, \partial_x \phi^{\text{VCB}}_{j'+1/2}(x) + \frac{K_{\uparrow\downarrow}}{K_{\uparrow\uparrow}} \sum_{j'\neq j} \text{sgn}(j'-j)\, \partial_x \widehat{\phi}_{j'\downarrow}(x). \tag{42}$$

Note that $\alpha_j(x)$ commutes with itself at different positions. Defining $\Delta\widetilde{\phi}_{j+1/2\downarrow}(x) = \widetilde{\phi}_{j+1\downarrow}(x) - \widetilde{\phi}_{j\downarrow}(x)$ and $\Delta\widetilde{\theta}^{\text{VCB}}_j(x) = \Delta\widetilde{\theta}^{\text{VCB}}_{j+1/2}(x) - \Delta\widetilde{\theta}^{\text{VCB}}_{j-1/2}(x)$, we find that the action transforms as

$$\mathcal{S} = \widetilde{\mathcal{S}}'_{2,\text{VCB}} + \widetilde{\mathcal{S}}'_{2,\downarrow} + \widetilde{\mathcal{S}}_{\text{SG}}, \tag{43}$$

$$\widetilde{\mathcal{S}}'_{2,\downarrow} = \int d^{1+1}x \sum_j \Big[ \frac{1}{\pi}(\partial_t \widetilde{\theta}_{j\downarrow} + q^*_\downarrow \mathcal{A}_{0,j})(\partial_x \widetilde{\phi}_{j\downarrow}) - \frac{\widehat{u}_\downarrow}{2\pi}(\partial_x \widetilde{\phi}_{j\downarrow})^2$$

$$- \frac{\widetilde{v}_\downarrow}{2\pi}(\partial_x \widetilde{\theta}_{j\downarrow}(x) - K_{\uparrow\downarrow}\alpha_j)^2 \Big], \tag{44}$$

$$\widetilde{\mathcal{S}}_{\text{SG}} = \int d^{1+1}x \sum_j (-g_{\uparrow,j+1/2}) \cos\Big(2\widetilde{\phi}^{\text{VCB}}_{j+1/2}(x) - q_\uparrow \ell_y \mathcal{A}_{y,j+1/2}\Big), \tag{45}$$

where we introduced $\widetilde{v}_\downarrow = \widehat{v}_\downarrow \left(K^{-1}_{\downarrow\downarrow}\right)^2$, and have

$$\widetilde{\mathcal{S}}'_{2,\text{VCB}} = \int d^{1+1}x \sum_j \Bigg[ \frac{1}{\pi}\Big(\partial_t \widetilde{\theta}^{\text{VCB}}_{j+1/2}\Big)(\partial_x \widetilde{\phi}^{\text{VCB}}_{j+1/2}) + \frac{1}{\pi}q_\uparrow \frac{\Delta\mathcal{A}_{0,j+1/2}}{2}(\partial_x \widetilde{\theta}^{\text{VCB}}_{j+1/2})$$

$$+ \frac{1}{\pi}q_\uparrow \mathcal{A}_{0,j}\frac{K_{\uparrow\downarrow}}{K_{\uparrow\uparrow}}\partial_x \widetilde{\phi}_{j\downarrow} - \frac{u_{\text{CB}} - K^2_{\uparrow\uparrow}v_{\text{CB}}}{8\pi}\Big(\partial_x \Delta\widetilde{\theta}^{\text{VCB}}_j(x) - 2\frac{K_{\uparrow\downarrow}}{K_{\uparrow\uparrow}}\partial_x \widetilde{\phi}_{j\downarrow}(x)\Big)^2$$

$$- \frac{v_{\text{CB}}K^2_{\uparrow\uparrow}}{2\pi}\Big(\partial_x \widetilde{\theta}^{\text{VCB}}_{j+1/2}(x) + S\alpha_{j+1/2}(x) + \frac{K_{\uparrow\downarrow}}{K_{\uparrow\uparrow}}\partial_x \frac{\Delta\widetilde{\phi}^\downarrow_{j+1/2}(x)}{2}\Big)^2$$

$$- \frac{v_{\text{CB}}}{2\pi}(\partial_x \widetilde{\phi}^{\text{VCB}}_{j+1/2}(x))^2 \Bigg]. \tag{46}$$

Due to the definition of $\alpha_j$, the action is now highly non-local. We can remedy this fact by the introduction of an emergent gauge field $\alpha_{j,\mu}$, for which we initially choose the gauge

$$\alpha_{j,2} = 0, \tag{47}$$

and enforce that

$$\alpha_{j,1} \overset{!}{=} -\frac{1}{K_{\uparrow\uparrow}} \sum_{j'} \text{sgn}(j'-j+1/2)\, \partial_x \widetilde{\phi}^{\text{VCB}}_{j'+1/2} + \frac{K_{\uparrow\downarrow}}{K_{\uparrow\uparrow}} \sum_{j'\neq j} \text{sgn}(j'-j)\, \partial_x \widetilde{\phi}_{j'\downarrow}(x). \tag{48}$$

We enforce the constraint via a Lagrange multiplier that we absorb into $\widetilde{\mathcal{S}}'_{2,\text{VCB}}$. After some algebra, the action can be brought to the form

$$\mathcal{S} = \widetilde{\mathcal{S}}_{2,\text{VCB}} + \widetilde{\mathcal{S}}_{2,\downarrow} + \widetilde{\mathcal{S}}_{\text{SG}}, \tag{49a}$$

$$\widetilde{\mathcal{S}}_{2,\downarrow} = \int d^{1+1}x \sum_j \Big[ \frac{1}{\pi}(\partial_t \widetilde{\theta}_{j\downarrow} + q_\downarrow \mathcal{A}_{0,j} - K_{\uparrow\downarrow}S\alpha_{j,0})(\partial_x \widetilde{\phi}_{j\downarrow})$$

$$- \frac{\widehat{u}_\downarrow}{2\pi}(\partial_x \widetilde{\phi}_{j\downarrow})^2 - \frac{\widetilde{v}_\downarrow}{2\pi}(\partial_x \widetilde{\theta}_{j\downarrow}(x) - K_{\uparrow\downarrow}\alpha_{j,1})^2 \Big], \tag{49b}$$

$$\widetilde{\mathcal{S}}_{\text{SG}} = \int d^{1+1}x \sum_j (-g_{\uparrow,j+1/2}) \cos\Big(2\widetilde{\phi}^{\text{VCB}}_{j+1/2}(x) - q_\uparrow \ell_y \mathcal{A}_{y,j+1/2}\Big), \tag{49c}$$

where

$$
\begin{aligned}
\widetilde{\mathcal{S}}_{2,\mathrm{VCB}} = \int d^{1+1}x \sum_j \Bigg[ & \frac{1}{\pi}\Big(\partial_t \widetilde{\theta}^{\mathrm{VCB}}_{j+1/2} + \alpha_{j+1/2,0}\Big)\big(\partial_x \widetilde{\phi}^{\mathrm{VCB}}_{j+1/2}\big) \\
& - \frac{u_{\mathrm{CB}} - K^2_{\uparrow\uparrow}\nu_{\mathrm{CB}}}{8\pi}\left(\partial_x \Delta\widetilde{\theta}^{\mathrm{VCB}}_{j} - 2\frac{K_{\uparrow\downarrow}}{K_{\uparrow\uparrow}}\partial_x \widetilde{\phi}_{j\downarrow}\right)^2 \\
& - \frac{\nu_{\mathrm{CB}}K^2_{\uparrow\uparrow}}{2\pi}\left(\partial_x \widetilde{\theta}^{\mathrm{VCB}}_{j+1/2} + S\alpha_{j+1/2,1} + \frac{K_{\uparrow\downarrow}}{K_{\uparrow\uparrow}}\partial_x \frac{\Delta\widetilde{\phi}^{\downarrow}_{j+1/2}}{2} - \frac{q_{\uparrow}}{2\nu_{\mathrm{CB}}K^2_{\uparrow\uparrow}}\Delta\mathcal{A}_{0,j+1/2}\right)^2 \\
& - \frac{\nu_{\mathrm{CB}}}{2\pi}(\partial_x \widetilde{\phi}^{\mathrm{VCB}}_{j+1/2})^2 - \frac{1}{2\pi}q_{\uparrow}\Delta\mathcal{A}_{0,j+1/2}S\alpha_{j+1/2,1} + \frac{q_{\uparrow}}{4\pi}\frac{K_{\uparrow\downarrow}}{K_{\uparrow\uparrow}}(\Delta^2\mathcal{A}_{0,j})\partial_x \widetilde{\phi}_{j\downarrow} \\
& + \frac{q^2_{\uparrow}}{8\pi\nu_{\mathrm{CB}}K^2_{\uparrow\uparrow}}\big(\Delta^2\mathcal{A}_{0,j}\big)\mathcal{A}_{0,j} + \frac{1}{2\pi}K_{\uparrow\uparrow}(\alpha_{j+1/2,0} - \alpha_{j-1/2,0})\alpha_{j,1}\Bigg].
\end{aligned}
\tag{50a}
$$

Here, we used that $\sum_j \Delta\mathcal{M}_{j+1/2}\Delta\mathcal{N}_{j+1/2} = -\sum_j(\Delta^2\mathcal{M}_j)\mathcal{N}_j$ denotes a a discrete version of a second derivative along $y$ (where we used $\Delta^2\mathcal{M}_j = \Delta\mathcal{M}_{j+1/2} - \Delta\mathcal{M}_{j-1/2}$).

## 5.1 Strong coupling phase of the sine-Gordon term

We are now interested in the strong coupling phase of the sine-Gordon term, for which the fields $\widetilde{\phi}^{\mathrm{VCB}}_{j+1/2}$ are pinned in a mean field approximation. We begin by integrating out the fields $\widetilde{\theta}^{\mathrm{VCB}}_{j+1/2}$ that are canonically conjugate to the pinned fields. To that end, we split the action $\widetilde{\mathcal{S}}_{2,\mathrm{VCB}}$ into parts containing $\widetilde{\theta}^{\mathrm{VCB}}_{j+1/2}$, and the rest as

$$
\widetilde{\mathcal{S}}_{2,\mathrm{VCB}} = \widetilde{\mathcal{S}}^{(1)}_{2,\mathrm{VCB}} + \widetilde{\mathcal{S}}^{(2)}_{2,\mathrm{VCB}},
\tag{51}
$$

with

$$
\begin{aligned}
\widetilde{\mathcal{S}}^{(1)}_{2,\mathrm{VCB}} = \int d^{1+1}x \sum_j \Bigg[ & \frac{1}{\pi}\left(\partial_t \widetilde{\phi}^{\mathrm{VCB}}_{j+1/2} - \frac{u_{\mathrm{CB}} - K^2_{\uparrow\uparrow}\nu_{\mathrm{CB}}}{2}\frac{K_{\uparrow\downarrow}}{K_{\uparrow\uparrow}}\partial_x \Delta\widetilde{\phi}^{\downarrow}_{j+1/2}\right)\big(\partial_x \widetilde{\theta}^{\mathrm{VCB}}_{j+1/2}\big) \\
& - \frac{u_{\mathrm{CB}} - K^2_{\uparrow\uparrow}\nu_{\mathrm{CB}}}{8\pi}\big(\partial_x \Delta\widetilde{\theta}^{\mathrm{VCB}}_{j}\big)^2 - \frac{u_{\mathrm{CB}} - K^2_{\uparrow\uparrow}\nu_{\mathrm{CB}}}{2\pi}\frac{K^2_{\uparrow\downarrow}}{K^2_{\uparrow\uparrow}}\big(\partial_x \widetilde{\phi}_{j\downarrow}\big)^2 \\
& - \frac{\nu_{\mathrm{CB}}K^2_{\uparrow\uparrow}}{2\pi}\left(\partial_x \widetilde{\theta}^{\mathrm{VCB}}_{j+1/2} + S\alpha_{j+1/2,1} + \frac{K_{\uparrow\downarrow}}{2K_{\uparrow\uparrow}}\partial_x \Delta\widetilde{\phi}^{\downarrow}_{j+1/2} - \frac{q_{\uparrow}}{2\nu_{\mathrm{CB}}K^2_{\uparrow\uparrow}}\Delta\mathcal{A}_{0,j+1/2}\right)^2\Bigg],
\end{aligned}
\tag{52}
$$

and

$$
\begin{aligned}
\widetilde{\mathcal{S}}^{(2)}_{2,\mathrm{VCB}} = \int d^{1+1}x \sum_j \Bigg[ & \frac{1}{\pi}\alpha_{j+1/2,0}(\partial_x \widetilde{\phi}^{\mathrm{VCB}}_{j+1/2}) \\
& - \frac{\nu_{\mathrm{CB}}}{2\pi}(\partial_x \widetilde{\phi}^{\mathrm{VCB}}_{j+1/2})^2 - \frac{1}{2\pi}q_{\uparrow}\Delta\mathcal{A}_{0,j+1/2}S\alpha_{j+1/2,1} + \frac{q_{\uparrow}}{4\pi}\frac{K_{\uparrow\downarrow}}{K_{\uparrow\uparrow}}(\Delta^2\mathcal{A}_{0,j})\partial_x \widetilde{\phi}_{j\downarrow} \\
& + \frac{q^2_{\uparrow}}{8\pi\nu_{\mathrm{CB}}K^2_{\uparrow\uparrow}}\big(\Delta^2\mathcal{A}_{0,j}\big)\mathcal{A}_{0,j} + \frac{1}{2\pi}K_{\uparrow\uparrow}(\alpha_{j+1/2,0} - \alpha_{j-1/2,0})\alpha_{j,1}\Bigg].
\end{aligned}
\tag{53}
$$

Next, we perform a discrete Fourier transformation along $y$. Using the definition $Q^2 = 2(1 - \cos(qa))$, the action $\widetilde{\mathcal{S}}^{(1)}_{2,\mathrm{VCB}}$ can be rewritten as

$$\widetilde{\mathcal{S}}_{2,\text{VCB}}^{(1)} = \int d^{1+1}x \sum_q \Bigg[ \frac{1}{2\pi} \left( \partial_t \widetilde{\phi}^{\text{VCB}}(q) - \frac{u_{\text{CB}} - K_{\uparrow\uparrow}^2 v_{\text{CB}}}{2} \frac{K_{\uparrow\downarrow}}{K_{\uparrow\uparrow}} \partial_x \Delta \widetilde{\phi}^{\downarrow}(q) \right) \left( \partial_x \widetilde{\theta}^{\text{VCB}}(-q) \right)$$

$$+ \frac{1}{2\pi} \left( \partial_t \widetilde{\phi}^{\text{VCB}}(-q) - \frac{u_{\text{CB}} - K_{\uparrow\uparrow}^2 v_{\text{CB}}}{2} \frac{K_{\uparrow\downarrow}}{K_{\uparrow\uparrow}} \partial_x \Delta \widetilde{\phi}^{\downarrow}(-q) \right) \left( \partial_x \widetilde{\theta}^{\text{VCB}}(q) \right)$$

$$- \frac{u_{\text{CB}} - K_{\uparrow\uparrow}^2 v_{\text{CB}}}{8\pi} Q^2 \left( \partial_x \widetilde{\theta}^{\text{VCB}}(q) \right) \left( \partial_x \widetilde{\theta}^{\text{VCB}}(-q) \right)$$

$$- \frac{u_{\text{CB}} - K_{\uparrow\uparrow}^2 v_{\text{CB}}}{2\pi} \frac{K_{\uparrow\downarrow}^2}{K_{\uparrow\uparrow}^2} \left( \partial_x \widetilde{\phi}^{\downarrow}(q) \right) \left( \partial_x \widetilde{\phi}^{\downarrow}(-q) \right)$$

$$- \frac{v_{\text{CB}} K_{\uparrow\uparrow}^2}{2\pi} \left( \partial_x \widetilde{\theta}^{\text{VCB}}(q) + S\alpha_1(q) + \frac{K_{\uparrow\downarrow}}{2K_{\uparrow\uparrow}} \partial_x \Delta \widetilde{\phi}^{\downarrow}(q) - \frac{q_\uparrow}{2v_{\text{CB}} K_{\uparrow\uparrow}^2} \Delta \mathcal{A}_0(q) \right)$$

$$\times \left( \partial_x \widetilde{\theta}^{\text{VCB}}(-q) + S\alpha_1(-q) + \frac{K_{\uparrow\downarrow}}{2K_{\uparrow\uparrow}} \partial_x \Delta \widetilde{\phi}^{\downarrow}(-q) - \frac{q_\uparrow}{2v_{\text{CB}} K_{\uparrow\uparrow}^2} \Delta \mathcal{A}_0(-q) \right) \Bigg]. \tag{54}$$

We are now in the position to integrate out the field $\widetilde{\theta}^{\text{VCB}}$. Here, the first approximation of our theory comes into play: we keep terms that at most contain two derivatives. We thus throw out all terms that contain, in total, three or more derivatives, including discrete derivatives in the $y$-direction. Next, we further simplify things in the strong-coupling phase of the sine-Gordon terms: there, we only keep terms involving the pinned field $\widetilde{\phi}^{\text{VCB}}$ and at most a first order of derivatives. As for the emergent gauge field, we focus on the leading terms. Those are of the form "current × gauge field". Since the current is a first derivative of the fields $\widetilde{\phi}^{\text{VCB}}$ and $\widetilde{\phi}^{\downarrow}$, this means that we only keep terms of the emergent gauge field that are at most of first order in derivatives. We then Fourier transform back to real space, and perform the same approximations for $\widetilde{\mathcal{S}}_{2,\text{VCB}}^{(2)}$. In addition, we drop a constant term $\left( \Delta^2 \mathcal{A}_{0,j} \right) \mathcal{A}_{0,j}$ that does not couple to the fermionic fields anymore. In the end, we can rewrite the action as

$$\mathcal{S} = \widetilde{\mathcal{S}}_{2,\text{VCB}}' + \widetilde{\mathcal{S}}_{2,\downarrow}' + \widetilde{\mathcal{S}}_{\text{SG}}, \tag{55a}$$

$$\widetilde{\mathcal{S}}_{2,\downarrow}' = \int d^{1+1}x \sum_j \Bigg[ \frac{1}{\pi} (\partial_t \widetilde{\theta}_{j\downarrow} + q_\downarrow \mathcal{A}_{0,j} - K_{\uparrow\downarrow} S\alpha_{j,0})(\partial_x \widetilde{\phi}_{j\downarrow})$$

$$- \frac{\widehat{u}_\downarrow}{2\pi} (\partial_x \widetilde{\phi}_{j\downarrow})^2 - \frac{\widetilde{v}_\downarrow}{2\pi} (\partial_x \widetilde{\theta}_{j\downarrow} - K_{\uparrow\downarrow} \alpha_{j,1})^2 - \frac{u_{\text{CB}} - K_{\uparrow\uparrow}^2 v_{\text{CB}}}{2\pi} \frac{K_{\uparrow\downarrow}^2}{K_{\uparrow\uparrow}^2} \left( \partial_x \widetilde{\phi}_j^{\downarrow} \right)^2 \Bigg], \tag{55b}$$

$$\widetilde{\mathcal{S}}_{\text{SG}}' = \int d^{1+1}x \sum_j (-g_{\uparrow,j+1/2}) \cos\left( 2\widetilde{\phi}_{j+1/2}^{\text{VCB}} - q_\uparrow \ell_y \mathcal{A}_{y,j+1/2} \right), \tag{55c}$$

$$\widetilde{\mathcal{S}}_{2,\text{VCB}}' \approx \int d^{1+1}x \sum_j \Bigg[ -\frac{1}{\pi} \left( \partial_t \widetilde{\phi}_{j+1/2}^{\text{VCB}} \right) S\alpha_{j+1/2,1} + \frac{1}{\pi} \alpha_{j+1/2,0} (\partial_x \widetilde{\phi}_{j+1/2}^{\text{VCB}})$$

$$- \frac{1}{2\pi} q_\uparrow \Delta \mathcal{A}_{0,j+1/2} S\alpha_{j+1/2,1} + \frac{1}{2\pi} K_{\uparrow\uparrow} (\alpha_{j+1/2,0} - \alpha_{j-1/2,0}) \alpha_{j,1} \Bigg]. \tag{55d}$$

In the strong-coupling phase of the sine-Gordon term, the field $\widetilde{\phi}_{j+1/2}^{\text{VCB}}(x)$ is pinned to

$$\widetilde{\phi}_{j+1/2}^{\text{VCB}}(x) = \frac{q_\uparrow \ell_y}{2} \mathcal{A}_{y,k+1/2}. \tag{56}$$

We thus perform the change of variables

$$\widetilde{\phi}_{j+1/2}^{\mathrm{VCB}}(x) = \bar{\phi}_{j+1/2}^{\mathrm{VCB}}(x) + \frac{q_\uparrow \ell_y}{2} \mathcal{A}_{y,k+1/2} = \bar{\phi}_{j+1/2}^{\mathrm{VCB}}(x) - \frac{q_\uparrow \ell_y}{2} \mathcal{A}_{2,k+1/2}. \tag{57}$$

In addition, we identify the quasiparticle density and quasiparticle current operators as

$$j_{\mathrm{qp},j+1/2}^0(x) = -\frac{1}{\pi \ell_y} \partial_x \bar{\phi}_{j+1/2}^{\mathrm{VCB}} \quad \text{and} \quad j_{\mathrm{qp},j+1/2}^1(x) = \frac{1}{\pi \ell_y} \partial_t \bar{\phi}_{j+1/2}^{\mathrm{VCB}}. \tag{58}$$

This brings the action to the form

$$\mathcal{S} = \bar{\mathcal{S}}_{2,\mathrm{VCB}} + \bar{\mathcal{S}}_{2,\downarrow} + \bar{\mathcal{S}}_{\mathrm{SG}}, \tag{59a}$$

$$\bar{\mathcal{S}}_{2,\downarrow} = \int d^{1+1}x \sum_j \left[ \frac{1}{\pi} (\partial_t \widetilde{\theta}_{j\downarrow} + q_\downarrow \mathcal{A}_{0,j} - K_{\uparrow\downarrow} S \alpha_{j,0})(\partial_x \widetilde{\phi}_{j\downarrow}) \right. \\ \left. - \frac{\widehat{u}_\downarrow}{2\pi} (\partial_x \widetilde{\phi}_{j\downarrow})^2 - \frac{\widetilde{v}_\downarrow}{2\pi} (\partial_x \widetilde{\theta}_{j\downarrow}(x) - K_{\uparrow\downarrow} \alpha_{j,1})^2 - \frac{u_{\mathrm{CB}} - K_{\uparrow\uparrow}^2 v_{\mathrm{CB}}}{2\pi} \frac{K_{\uparrow\downarrow}^2}{K_{\uparrow\uparrow}^2} \left( \partial_x \widetilde{\phi}_j^\downarrow \right)^2 \right], \tag{59b}$$

$$\bar{\mathcal{S}}_{\mathrm{SG}} = \int d^{1+1}x \sum_j (-g_{\uparrow,j+1/2}) \cos\!\left( 2\bar{\phi}_{j+1/2}^{\mathrm{VCB}}(x) \right), \tag{59c}$$

$$\bar{\mathcal{S}}_{2,\mathrm{VCB}} \approx \int d^{1+1}x \sum_j \left[ -\ell_y j_{\mathrm{qp},k+1/2}^1 S \alpha_{j+1/2,1} + \frac{q_\uparrow \ell_y}{2\pi} S \alpha_{j+1/2,1} \partial_t \mathcal{A}_{2,k+1/2} \right. \\ - \frac{q_\uparrow a}{2\pi} \alpha_{j+1/2,0} \partial_x \mathcal{A}_{2,k+1/2} - \ell_y j_{\mathrm{qp},k+1/2}^0 \alpha_{j+1/2,0} \\ \left. - \frac{1}{2\pi} q_\uparrow S \alpha_{j+1/2,1} \Delta \mathcal{A}_{0,j+1/2} + \frac{1}{2\pi} K_{\uparrow\uparrow} \alpha_{j,1} \Delta \alpha_{j,0} \right]. \tag{59d}$$

We recognize that $\bar{\mathcal{S}}_{2,\mathrm{VCB}}$ is a discretized version of

$$\bar{\mathcal{S}}_{2,\mathrm{VCB}}^{\mathrm{cont.}} = \int d^{2+1}x \left[ -j_{\mathrm{qp}}^\mu \alpha_\mu - \frac{q_\uparrow}{2\pi} \epsilon^{\mu\nu\lambda} \alpha_\mu \partial_\nu \mathcal{A}_\lambda + \frac{1}{4\pi} K_{\uparrow\uparrow} \epsilon^{\mu\nu\lambda} \alpha_\mu \partial_\nu \alpha_\lambda \right] \tag{60}$$

in the gauges $\mathcal{A}_1 = 0$ and $\alpha_2 = 0$. In total, we thus find that the gauge-invariant extension of the low-energy action in the continuum limit along the $y$-direction reads

$$\mathcal{S} \approx \bar{\mathcal{S}}_{2,\downarrow} + \bar{\mathcal{S}}_{2,\mathrm{VCB}}^{\mathrm{cont.}} + \bar{\mathcal{S}}_{\mathrm{SG}}, \tag{61a}$$

$$\bar{\mathcal{S}}_{2,\downarrow} = \int d^{1+1}x \sum_j \left[ \frac{1}{\pi} (\partial_t \widetilde{\theta}_{j\downarrow} + q_\downarrow \mathcal{A}_{0,j} - K_{\uparrow\downarrow} S \alpha_{j,0})(\partial_x \widetilde{\phi}_{j\downarrow}) \right. \\ \left. - \frac{\bar{u}_\downarrow}{2\pi} (\partial_x \widetilde{\phi}_{j\downarrow})^2 - \frac{\bar{v}_\downarrow}{2\pi} (\partial_x \widetilde{\theta}_{j\downarrow}(x) - K_{\uparrow\downarrow} \alpha_{j,1})^2 \right], \tag{61b}$$

$$\bar{\mathcal{S}}_{2,\mathrm{VCB}}^{\mathrm{cont.}} = \int d^{2+1}x \left[ -j_{\mathrm{qp}}^\mu \alpha_\mu - \frac{q_\uparrow}{2\pi} \epsilon^{\mu\nu\lambda} \alpha_\mu \partial_\nu \mathcal{A}_\lambda + \frac{1}{4\pi} K_{\uparrow\uparrow} \epsilon^{\mu\nu\lambda} \alpha_\mu \partial_\nu \alpha_\lambda \right], \tag{61c}$$

$$\bar{\mathcal{S}}_{\mathrm{SG}} = \int d^{1+1}x \sum_j (-g_{\uparrow,j+1/2}) \cos\!\left( 2\bar{\phi}_{j+1/2}^{\mathrm{VCB}}(x) \right), \tag{61d}$$

where

$$\bar{u}_\downarrow = \widehat{u}_\downarrow + \frac{K_{\uparrow\downarrow}^2}{K_{\uparrow\uparrow}^2} \left( u_{\mathrm{CB}} - K_{\uparrow\uparrow}^2 v_{\mathrm{CB}} \right) \quad \text{and} \quad \bar{v}_\downarrow = \widetilde{v}_\downarrow. \tag{62}$$

# 6 The gapless sector: electrons glued to Laughlin quasi-holes

In Eq. (61c), the charge of the gapless quasiparticles is somewhat obscured by the fact that they couple both directly to the electromagnetic gauge field $\mathcal{A}$ with a charge $q_\downarrow$, and to the emergent gauge field $\alpha$, which in turn again couples to the electromagnetic gauge field. The coupling between the emergent gauge field and the gapless quasiparticles therefore mediates an additional channel by which the quasiparticles couples to the electromagnetic field, which in turn modifies their electric. One way to determine the total electric charge of the quasiparticles is thus to shift the gauge field as $\beta_\mu = \alpha_\mu - \mathcal{A}_\mu \frac{q_\uparrow}{K_{\uparrow\uparrow}}$. Plugging this into the action, we obtain

$$
\mathcal{S} = S_\downarrow^{2+1} - \int d^{2+1}x \, j^\mu \left( q_\downarrow^* \mathcal{A}_\mu - K_{\uparrow\downarrow} \beta_\mu \right) + \int d^{2+1}x \left[ -j_{\rm qp}^\mu \beta_\mu + \frac{1}{4\pi} K_{\uparrow\uparrow} \epsilon^{\mu\nu\lambda} \beta_\mu \, \partial_\nu \beta_\lambda \right]
$$
$$
+ \int d^{2+1}x \left[ -\frac{q_\uparrow}{K_{\uparrow\uparrow}} j_{\rm qp}^\mu \mathcal{A}_\mu - \frac{q_\uparrow^2}{4\pi} \frac{1}{K_{\uparrow\uparrow}} \epsilon^{\mu\nu\lambda} \mathcal{A}_\mu \, \partial_\nu \mathcal{A}_\lambda \right], \tag{63}
$$

where $q_\downarrow^* = q_\downarrow - \frac{K_{\uparrow\downarrow}}{K_{\uparrow\uparrow}} q_\uparrow$ is the effective charge already introduced above. At the same time, we see that quasiparticles in the gapped sector couple to the electromagnetic gauge field with a charge $q_\uparrow^* = q_\uparrow / K_{\uparrow\uparrow}$.

## 6.1 Charge operator

One of the benefits of our coupled-wire construction is that it provides us with microscopic expressions for all quantities. We can for example find an alternative viewpoint for the fractional effective charges $q^*$ by rewriting the charge density operator using the transformed bosonized fields. In terms of the original electronic fields, the total charge density is given by

$$
\rho(x) = -\frac{1}{\pi\ell_y} \sum_{j,\sigma} q_\sigma \partial_x \phi_{j\sigma}(x) = -\frac{1}{2\pi\ell_y} \sum_\sigma q_\sigma \partial_x \left( \Phi_{Rj\sigma}(x) - \Phi_{Lj\sigma}(x) \right)
$$
$$
= -\frac{1}{\pi\ell_y} \sum_j \left( \frac{q_\uparrow}{K_{\uparrow\uparrow}} \partial_x \left( \frac{\widetilde{\Phi}_{Rj\uparrow}(x) - \widetilde{\Phi}_{Lj\uparrow}(x)}{2} \right) \right.
$$
$$
\left. + \left( q_\downarrow - \frac{K_{\uparrow\downarrow}}{K_{\uparrow\uparrow}} q_\uparrow \right) \partial_x \left( \frac{\Theta_{Rj\downarrow}(x) - \Theta_{Lj\downarrow}(x)}{2} \right) \right). \tag{64}
$$

Next, we use that $\Theta_{Rj\downarrow} - \Theta_{Lj\downarrow} = \widetilde{\Theta}'_{Rj\downarrow} - \widetilde{\Theta}'_{Lj\downarrow}$ and shift the summation for the first term to obtain

$$
\rho(x) = -\frac{1}{2\pi\ell_y} \partial_x \left( \frac{q_\uparrow}{K_{\uparrow\uparrow}} \sum_j \left[ \widetilde{\Phi}_{Rj\uparrow}(x) - \widetilde{\Phi}_{Lj+1\uparrow}(x) \right] + q_\downarrow^* \sum_j \left[ \widetilde{\Theta}'_{Rj\downarrow}(x) - \widetilde{\Theta}'_{Lj\downarrow}(x) \right] \right)
$$
$$
= -\frac{1}{2\pi\ell_y} \partial_x \left( \frac{q_\uparrow}{K_{\uparrow\uparrow}} \sum_j 2 \, \widetilde{\phi}_{j+1/2}^{\rm VCB}(x) + q_\downarrow^* \sum_j \left[ \widetilde{\Theta}'_{Rj\downarrow}(x) - \widetilde{\Theta}'_{Lj\downarrow}(x) \right] \right). \tag{65}
$$

In the gapped sector, a quasiparticle is associated with a $2\pi$-kink in one of the sine-Gordon terms. Eq. (65) shows that such a kink carries a charge $q_\uparrow^* = q_\uparrow / K_{\uparrow\uparrow}$. Furthermore, a quasiparticle in the gapless sector is created by an operator $\sim \exp\left\{ \widetilde{\Theta}'_{rj\downarrow} \right\}$. Commuting this operator with the charge density operator shows that such a quasiparticle carries a charge $q_\downarrow^*$, in agreement with the above result.

## 6.2 Electron creation and annihilation operators

It is also interesting to re-express the electronic creation and annihilation operators in the basis of the new fields. With some straightforward but tedious algebra, one finds

$$\Phi_{rj\downarrow} = \hat{r}\,\phi_{j\downarrow} - \theta_{j\downarrow} = \hat{r}\,\widetilde{\phi}_j^{\downarrow} - \widetilde{\theta}_j^{\downarrow} - n_1\,\widetilde{\theta}_{j+1/2}^{\mathrm{VCB}} - n_2\,\widetilde{\theta}_{j-1/2}^{\mathrm{VCB}}\,. \tag{66}$$

Since an exponential of $\hat{r}\,\widetilde{\phi}_j^{\downarrow} - \widetilde{\theta}_j^{\downarrow}$ creates a quasiparticle in the gapless sector, while an exponential of $\widetilde{\theta}_{j+1/2}^{\mathrm{VCB}}$ creates a quasiparticle in the gapped sector, we find that creating an electron is equivalent to creating a quasiparticle in the gapless sector and $n_1 + n_2 = K_{\downarrow\uparrow}$ quasiparticles in the gapped sector. Since an electron carries charge $q_{\downarrow}$, and because quasiparticles in the gapped sector have charge $q_{\uparrow}/K_{\uparrow\uparrow}$, the quasiparticles in the gapless sector must carry charge $q_{\downarrow}^* = q_{\downarrow} - K_{\downarrow\uparrow} \times \frac{q_{\uparrow}}{K_{\uparrow\uparrow}}$, in agreement with the earlier findings.

## 6.3 Sine-Gordon term as glue between electrons and fractional quasiparticles

Our coupled-wire construction identifies the sine-Gordon term in Eq. (19) as the glue between the electrons and Laughlin-like quasiparticles that compose the gapless quasiparticles. Namely, the coupled-wire construction of Laughlin states in Ref. [1] shows that the number density of Laughlin quasiparticles in a $1/K_{\uparrow\uparrow}$-state is

$$\rho_{j+1/2}^{1/K_{\uparrow\uparrow}} = -\partial_x\left(\widetilde{\Phi}_{Rj\uparrow}^{n_1=n_2=0} - \widetilde{\Phi}_{Lj+1\uparrow}^{n_1=n_2=0}\right)/2\pi\ell_y, \tag{67}$$

where $\widetilde{\Phi}_{rj\uparrow}^{n_1=n_2=0}$ is defined by Eq. (20) with $n_1 = n_2 = 0$. Combined with the density of electrons in the bottom layer, $\rho_{j\downarrow} = \sum_r \rho_{rj\downarrow}$, we find that

$$-\partial_x \frac{\widetilde{\Phi}_{Rj\uparrow} - \widetilde{\Phi}_{Lj+1\uparrow}}{2\pi\ell_y} = \rho_{j+1/2}^{1/K_{\uparrow\uparrow}} + n_1\,\rho_{j\downarrow} + n_2\,\rho_{j+1\downarrow}. \tag{68}$$

In its strong coupling phase, $H_{g\uparrow}$ wants to pin $\widetilde{\Phi}_{Rj\sigma} - \widetilde{\Phi}_{Lj+1\sigma}$ to a constant value. A change of the bottom layer electron density $\rho_{j\downarrow} \to \rho_{j\downarrow} + 1$ thus constitutes an excitation out of the ground state of $H_{g\uparrow}$ unless we adapt the Laughlin-like quasiparticle density in the top layer as $\rho_{j+1/2}^{1/K_{\uparrow\uparrow}} \to \rho_{j+1/2}^{1/K_{\uparrow\uparrow}} - n_1$ and $\rho_{j-1/2}^{1/K_{\uparrow\uparrow}} \to \rho_{j-1/2}^{1/K_{\uparrow\uparrow}} - n_2$. Gapless excitations of a semi-quantized quantum Hall state hence correspond to electrons in the bottom layer glued to $n_1 + n_2 = K_{\uparrow\downarrow}$ Laughlin-like quasi-holes in the top layer.

# 7 Refermionization of the gapless sector and discussion of the low-energy action

Since the fields $\widetilde{\phi}_{j\downarrow}$ and $\widetilde{\theta}_{j\downarrow}$ obey the canonical commutator for the bosonized field of a fermionic theory, we can refermionize the gapless sector by introducing the composite quasiparticle fields $\psi_{rj}^{\mathrm{CQP}} = \exp\{-i(\hat{r}\,\widetilde{\phi}_{j\downarrow} - \widetilde{\theta}_{j\downarrow})\}$ [12]. The universal topological low-energy theory corresponds the continuum limit of our coupled-wire construction in $y$-direction, in which differences of fields in adjacent wires are replaced by a $y$-derivative. Shifting the gauge field by introducing $\beta_\mu = \alpha_\mu - \mathcal{A}_\mu \frac{q_{\uparrow}}{K_{\uparrow\uparrow}}$, we obtain the final form of the action as

$$S = \int d^{2+1}x \left( \sum_r \mathcal{L}_r^{\text{CQP}} + \mathcal{L}_{\text{int}}^{\text{CQP}} + \mathcal{L}_{\text{CS}} + \mathcal{L}_{\mathcal{A}} \right) + \sum_j \int d^{1+1}x \, \mathcal{L}_{g\uparrow,j+1/2} \text{ with}$$

$$\mathcal{L}_r^{\text{CQP}} = \bar{\psi}_r^{\text{CQP}} \, \widetilde{\mathcal{E}}_r \left( -i\partial_\mu + q_\downarrow^* \mathcal{A}_\mu - K_{\uparrow\downarrow} \beta_\mu \right) \psi_r^{\text{CQP}} \tag{69a}$$

$$\mathcal{L}_{\text{CS}} = -j_{\text{qp}}^\mu \beta_\mu + \frac{1}{4\pi} K_{\uparrow\uparrow} \epsilon^{\mu\nu\lambda} \beta_\mu \partial_\nu \beta_\lambda, \tag{69b}$$

$$\mathcal{L}_{\mathcal{A}} = -\frac{q_\uparrow}{K_{\uparrow\uparrow}} j_{\text{qp}}^\mu \mathcal{A}_\mu - \frac{q_\uparrow^2}{4\pi} \frac{1}{K_{\uparrow\uparrow}} \epsilon^{\mu\nu\lambda} \mathcal{A}_\mu \partial_\nu \mathcal{A}_\lambda. \tag{69c}$$

This form of the action is equivalent to Eq. (12) obtained in a heuristic continuum theory. The gapless ↓-sector is described by a non-universal Lagrangian $\mathcal{L}^{\text{CQP}} = \sum_r \mathcal{L}_r^{\text{CQP}} + \mathcal{L}_{\text{int}}^{\text{CQP}}$ for the gapless composite quasiparticles $\psi_r^{\text{CQP}}$, in which the single-particle energy $\widetilde{\mathcal{E}}_r$ and the interactions described by $\mathcal{L}_{\text{int}}^{\text{CQP}}$ depend on system-specific details. The gapless composite quasiparticles are minimally coupled both to the electromagnetic potential with a fractional charge

$$q_\downarrow^* = q_\downarrow - \frac{K_{\uparrow\downarrow}}{K_{\uparrow\uparrow}} q_\uparrow, \tag{70}$$

and to the emergent gauge field $\beta$. The emergent gauge field is governed by a Chern-Simons theory $\mathcal{L}_{\text{CS}}$. It physically derives from a Hall effect in the top layer, whose gapless edge states carry the background Hall conductance encoded in $\mathcal{L}_{\mathcal{A}}$. This Hall effect and the charges of the gapped quasiparticles match a Laughlin state at a filling of $1/K_{\uparrow\uparrow}$. Finally, the sine-Gordon terms in their strong-coupling phase, contained in $\mathcal{L}_{g\uparrow,j+1/2} = (-g_{\uparrow,j+1/2}) \cos\left(2 \widetilde{\phi}_{j+1/2}^{\text{VCB}} + q_\uparrow \ell_2 \mathcal{A}_{2,j+1/2}\right)$, encode the energy cost for quasiparticles in the gapped sector.

In conclusion, we find that the gapless composite quasiparticles carry a fractional charge, and should be viewed as forming a liquid of anyons rather than fermions since they couple to the emergent gauge field.

## 8 Electromagnetic Response

Experimentally, semi-quantized quantum Hall states are most easily deteced by their electromagnetic responses, in particular the Hall and Hall drag responses. To calculate the latter, we need to keep track of charges and electric fields in both layer separately. Since the electromagnetic fields couple to the quasiparticles via the charges $q_\sigma$, we can simply replace $q_\sigma \mathcal{A}_\mu \to q_\sigma \mathcal{A}_{\mu,\sigma}$ to resolve the fields in each layer. The background Hall current in the gapped ↑-sector follows from

$$j_{\uparrow,\text{Hall}}^\mu = -\frac{\delta}{\delta \mathcal{A}_{\mu,\uparrow}} \int d^{2+1}x \left( -\frac{1}{4\pi} \frac{q_\uparrow^2}{K_{\uparrow\uparrow}} \epsilon^{\mu\nu\lambda} \mathcal{A}_{\mu,\uparrow} \partial_\nu \mathcal{A}_{\lambda\uparrow} \right) = \frac{1}{2\pi} \frac{q_\uparrow^2}{K_{\uparrow\uparrow}} \epsilon^{\mu\nu\lambda} \partial_\nu \mathcal{A}_{\lambda,\uparrow}. \tag{71}$$

The gapless sector is described by composite fermions that couple to the combination of fields $q_\downarrow^* \mathcal{A}_\mu \to q_\downarrow \mathcal{A}_{\mu,\downarrow} - \frac{K_{\uparrow\downarrow}}{K_{\uparrow\uparrow}} q_\uparrow \mathcal{A}_{\mu,\uparrow}$. We can thus split the current of the composite quasiparticles into its contributions in the two layers as

$$j^\mu_{\text{CQP},\downarrow} = -\frac{\delta}{\delta A_{\mu,\downarrow}} \underbrace{\int d^{2+1}x \left( \ldots - \bar{\Psi} \mathcal{E} \left[ -i\partial_\mu + q^*_\downarrow \mathcal{A}_\mu - K_{\uparrow\downarrow} \beta_\mu \right] \Psi \right)}_{=\mathcal{S}_{\text{CQP}}}$$

$$= -\frac{\delta \mathcal{S}_{\text{CQP}}}{\delta(q^*_\downarrow \mathcal{A}_\mu)} \frac{d(q^*_\downarrow \mathcal{A}_\mu)}{d\mathcal{A}_{\mu,\downarrow}} = -\frac{\delta \mathcal{S}_{\text{CQP}}}{\delta(q^*_\downarrow \mathcal{A}_\mu)} q_\downarrow \tag{72}$$

and, similarly,

$$j^\mu_{\text{CQP},\uparrow} = \frac{\delta \mathcal{S}_{\text{CQP}}}{\delta(q^*_\downarrow \mathcal{A}_\mu)} \frac{K_{\uparrow\downarrow}}{K_{\uparrow\uparrow}} q_\uparrow. \tag{73}$$

Defining $j^\mu_{\text{CQP},0} = -q_\downarrow \frac{\delta \mathcal{S}_{\text{CQP}}}{\delta(q^*_\downarrow \mathcal{A}_\mu)}$, we have $j^\mu_{\text{CQP},\downarrow} = j^\mu_{\text{CQP},0}$ and $j^\mu_{\text{CQP},\uparrow} = -\frac{q_\uparrow}{q_\downarrow} \frac{K_{\uparrow\downarrow}}{K_{\uparrow\uparrow}} j^\mu_{\text{CQP},0}$.

## 8.1 Current responses if one layer is driven and the other layer is disconnected: perfect drag

We now focus on the electric response of a semi-quantized quantum Hall state in an experiment in which both layers can be independently driven and measured, similar to the setups reported in Refs. [7, 8]. The total current flowing in the top layer is composed of the background Hall response of the gapped sector, plus a contribution of the gapless composite quasiparticles. The current in the bottom layer, on the other hand, is carried only by the gapless composite quasiparticles.

Since the composite quasiparticles in the gapless $\downarrow$-sector couple to the electromagnetic potential via the combination of fields $q_\downarrow \mathcal{A}_{\mu,\downarrow} - \frac{K_{\uparrow\downarrow}}{K_{\uparrow\uparrow}} q_\uparrow \mathcal{A}_{\mu,\uparrow}$, the $m$-th component of the current of the composite quasiparticles (with $m = x, y$, not $\mu = 1, 2$) will linearly respond to electric fields as

$$j^m_{\text{CQP},0} = \sigma^{mn}_{\text{CQP}} \left( E_{n,\downarrow} - \frac{q_\uparrow}{q_\downarrow} \frac{K_{\uparrow\downarrow}}{K_{\uparrow\uparrow}} E_{n,\uparrow} \right), \tag{74}$$

where $E_{n,\sigma}$ is the $n$-th component of the electric field in layer $\sigma$ (with $n = x, y$), and where the Einstein sum convention is understood. The conductivity $\sigma^{mn}_{\text{CQP}}$ is a non-universal function that depends on the scattering mechanisms in the gapless layer. Combining our above results, we find

$$j_{\uparrow,m} = \frac{1}{2\pi} \frac{q^2_\uparrow}{K_{\uparrow\uparrow}} \left( \delta_{my} E_{x,\uparrow} - \delta_{mx} E_{y,\uparrow} \right) - \sigma^{mn}_{\text{CQP}} \frac{q_\uparrow}{q_\downarrow} \frac{K_{\uparrow\downarrow}}{K_{\uparrow\uparrow}} \left( E_{n,\downarrow} - \frac{q_\uparrow}{q_\downarrow} \frac{K_{\uparrow\downarrow}}{K_{\uparrow\uparrow}} E_{n,\uparrow} \right), \tag{75a}$$

$$j_{\downarrow,m} = \sigma^{mn}_{\text{CQP}} \left( E_{n,\downarrow} - \frac{q_\uparrow}{q_\downarrow} \frac{K_{\uparrow\downarrow}}{K_{\uparrow\uparrow}} E_{n,\uparrow} \right). \tag{75b}$$

**Driving the top layer.** At first, we analyze a situation in which the top layer is driven, while the bottom layer is disconnected. In the steady state, there hence cannot be any current flowing in the bottom layer. From $j_{\downarrow,m} = 0$, we find that for $\sigma^{mn}_{\text{CQP}} \neq 0$, this implies

$$E_{n,\downarrow} = \frac{q_\uparrow}{q_\downarrow} \frac{K_{\uparrow\downarrow}}{K_{\uparrow\uparrow}} E_{n,\uparrow}. \tag{76}$$

We interpret this result as follows: if an electric field $E_{n,\uparrow}$ is applied to the top layer, the part of the gapless composite quasiparticles in the gapless $\downarrow$-sector that lives in the top layer, i.e. the

Laughlin quasiparticles glued to the electrons in the bottom layer, is initially accelerated by this field. Since these Laughlin quasiparticles are glued to the electrons in the ↓-layer, the latter will also start to move. As a result, an initial current will flow in the bottom layer. This layer is, however, electrically disconnected by assumption. The current in the bottom layer thus eventually hits the edge, and leads to a charge build-up. The corresponding electric field in the bottom layer opposes the current flow there. The steady state is reached if the net field felt by the composite quasiparticles (composed of the electric field in the bottom layer felt by the electrons there, and the field in the top layer felt by the Laughlin quasiparticles there) vanishes. In this situation, the current in the top layer is then given by

$$j_{\uparrow,m} = \frac{1}{2\pi} \frac{q_\uparrow^2}{K_{\uparrow\uparrow}} \left( \delta_{my} E_{x,\uparrow} - \delta_{mx} E_{y,\uparrow} \right), \tag{77}$$

and thus a pure Hall current. For $\boldsymbol{E} = E_x \hat{\boldsymbol{e}}_{\boldsymbol{x}}$, we thus find that the Hall resistance in the top layer and the Hall drag resistance in the bottom layer, resulting from driving the top layer with a disconnected bottom layer are

$$R_H^{\uparrow,\text{drive}} = \frac{E_{x,\uparrow}}{j_\uparrow^y} = \frac{h}{q_\uparrow^2} K_{\uparrow\uparrow} \quad \text{and} \quad R_H^{\downarrow,\text{drag}} = \frac{E_{x,\downarrow}}{j_\uparrow^y} = \frac{h}{q_\uparrow q_\downarrow} K_{\uparrow\downarrow}. \tag{78}$$

This response is in line with the argument put forward in Ref. [7].

**Driving the bottom layer.** If the bottom layer is driven, a current

$$j_{\downarrow,m} = \sigma_{\text{CQP}}^{mn} \left( E_{n,\downarrow} - \frac{q_\uparrow}{q_\downarrow} \frac{K_{\uparrow\downarrow}}{K_{\uparrow\uparrow}} E_{n,\uparrow} \right) \tag{79}$$

will flow there (as mentioned above, the current has some non-universal value reflecting the microscopic scattering mechanisms in the gapless sub-sector). If the top layer is electrically disconnected, we must have $j_{\uparrow,m} = 0$. This condition requires an induced Hall voltage in the top layer. Namely, the absence of a current in the top layer implies that

$$\frac{1}{2\pi} \frac{q_\uparrow^2}{K_{\uparrow\uparrow}} \left( \delta_{my} E_{x,\uparrow} - \delta_{mx} E_{y,\uparrow} \right) = \frac{q_\uparrow}{q_\downarrow} \frac{K_{\uparrow\downarrow}}{K_{\uparrow\uparrow}} j_{\downarrow,m}. \tag{80}$$

Let us for concreteness analyze the case that the current in the bottom layer flows in $y$-direction according to

$$j_\downarrow^y = \sigma_{\text{CQP}}^{ym} \left( E_{m,\downarrow} - \frac{q_\uparrow}{q_\downarrow} \frac{K_{\uparrow\downarrow}}{K_{\uparrow\uparrow}} E_{m,\uparrow} \right). \tag{81}$$

This implies that $\frac{1}{2\pi} \frac{q_\uparrow^2}{K_{\uparrow\uparrow}} E_{x,\uparrow} = \frac{q_\uparrow}{q_\downarrow} \frac{K_{\uparrow\downarrow}}{K_{\uparrow\uparrow}} j_{\downarrow,y}$. We can thus again define a Hall drag resistance, now for the top layer, which equals

$$R_H^{\uparrow,\text{drag}} = \frac{E_{x,\uparrow}}{j_\downarrow^y} = \frac{h}{q_\uparrow q_\downarrow} K_{\uparrow\downarrow}. \tag{82}$$

This response is also in agreement with the arguments presented in Ref. [7]: a current in the bottom layer necessarily is a current of the gapless composite quasiparticles (at least in the linear response regime). Because the quasiparticles are composed of electrons in the bottom

layer and Laughlin quasiparticles in the top layer, this current of quasiparticles implies a current in the top layer as well. However, the top layer is electrically disconnected, such that it cannot carry a net current. The bulk current due to the quasiparticle flow must hence be compensated by a Hall current flowing at the edges of the top layer. In simple pictures, when the composite quasiparticle hits the edge, it splits up: the electron in the bottom layer hops out of the sample and leads to a net current flow. The Laughlin quasiparticles in the top layer enter the gapless edge channel, and along the edge flow back to the other side of the sample. This leads to a population imbalance of the edge channels on the two sides, which in turn translates to an electrochemical potential difference transverse to the edge channels, and thus a Hall voltage in the top layer.

## 8.2 Discriminating semi-quantized quantum Hall states from Halperin bilayer states

Halperin bilayer states also feature quantized Hall and Hall drag resistances. In contrast to semi-quantized quantum Hall states, however, both layer of a Halperin bilayer state are fully gapped. Therefore, the longitudinal resistance of both layers always vanishes, and the Hall resistance of both layer is fully quantized. In contrast, semi-quantized quantum Hall states have one layer with a non-universal longitudinal resistance and non-quantized Hall resistance (in our example the bottom layer). A transport experiment can thus distinguish between semi-quantized quantum Hall states and Halperin bilayer states by measuring the longitudinal and Hall resistances of each layer separately, as was done in Ref. [7].

# 9 Beyond Hall drag measurements: spectral probes, non-Fermi liquid physics, and fully gapped daughter states

We finally discuss additional experimental properties of semi-quantized quantum Hall states that should be addressed by future experiments.

## 9.1 Gapped electronic spectrum

Since the gapless quasiparticles are composite objects, electronic spectral probes such as tunnelling spectroscopy exhibit a full gap. The size of the gap is set by the glueing energy scale $H_{g\uparrow}$ that needs to be overcome when extracting an electron from the sample. Formally, the above exact transformations show that the original electronic operator can be written as exponentials of $\Phi_{rj\downarrow} = (r\,\widetilde{\phi}_j^\downarrow - \widetilde{\theta}_j^\downarrow) - n_1\,\widetilde{\theta}_{j+1/2}^{\text{VCB}} - n_2\,\widetilde{\theta}_{j-1/2}^{\text{VCB}}$, where an exponential of $\widetilde{\theta}_{j+1/2}^{\text{VCB}}$ indeed creates a gapped quasiparticle.

## 9.2 Fully gapped daughter states with anyonic excitations

Fully gapped states deriving from semi-quantized quantum Hall states also show tantalizing properties. Consider for example a charge-density wave (CDW) gapping the composite quasiparticles. The non-trivial character of semi-quantized quantum Hall states is handed down to quasiparticles above the CDW gap in two ways: they carry a fractional charge $q_\downarrow^*$, and exhibit anyonic braiding. Namely, the coupling to the emergent gauge field results in a phase $\varphi_{\downarrow\downarrow} = 2\pi \frac{K_{\uparrow\downarrow}^2}{K_{\uparrow\uparrow}}$ for a braid of two CDW-quasiparticles, while a braid between a CDW-quasiparticle and a quasiparticle in the $\uparrow$-sector yields a phase $\varphi_{\downarrow\uparrow} = -2\pi \frac{K_{\uparrow\downarrow}}{K_{\uparrow\uparrow}}$, in agreement with the physical picture of electrons glued to Laughlin-like quasiholes (see the Appendix for further details of a CDW state).

The gapless composite quasiparticles in general feel a net magnetic field composed of the external field and the emergent gauge field. In addition to the constraint $K_{\uparrow\uparrow} \, \nu_\uparrow + K_{\uparrow\downarrow} \, \nu_\downarrow = 1$ established in Eq. 18, the filling factors also respect $K_{\downarrow\downarrow} \, \nu_\downarrow + K_{\uparrow\downarrow} \, \nu_\uparrow = 1$, then the system can in principle form a fully gapped $(K_{\uparrow\uparrow}, K_{\downarrow\downarrow}, K_{\uparrow\downarrow})$-Halperin bilayer state. This happens (for $q_\downarrow = q_\uparrow = e$) at filling factors

$$\nu_\uparrow = \frac{K_{\downarrow\downarrow} - K_{\uparrow\downarrow}}{K_{\uparrow\uparrow} K_{\downarrow\downarrow} - K_{\uparrow\downarrow}^2} \quad \text{and} \quad \nu_\downarrow = \frac{K_{\uparrow\uparrow} - K_{\uparrow\downarrow}}{K_{\uparrow\uparrow} K_{\downarrow\downarrow} - K_{\uparrow\downarrow}^2}. \tag{83}$$

In the present formulation, these states are $1/K_{\downarrow\downarrow}$-Laughlin states for the composite quasiparticles.

## 9.3 Non-Fermi liquid physics

The action in Eq. (69) is closely related to the composite Fermi liquid theory proposed by Halperin, Lee and Read (HLR) [9]. In this context, interactions between gapless composite quasiparticles mediated by an emergent gauge field as in $\mathcal{L}_{CS}$ were studied in detail. It was shown that the combination of the Chern-Simons gauge-field mediated interaction and additional density-density interactions of short range can lead to non-Fermi liquid physics. This will for example be heralded by a non-Fermi liquid scaling of the resistivity with temperature [9–11]. We expect the same to hold for semi-quantized quantum Hall states. In the absence of quasiparticles in the gapped ↑-sector, the action of a semi-quantized quantum Hall state is of the same form as the HLR-action, modulo a modified prefactor of the Chern-Simons term and the modified charge $q^*$ of the gapless quasiparticles (the effective magnetic field for the gapless composite quasiparticles vanishes at $\nu_\downarrow = \frac{1}{K_{\uparrow\downarrow}} - \frac{K_{\uparrow\uparrow}}{K_{\uparrow\downarrow}^2}$ and $\nu_\uparrow = 1/K_{\uparrow\downarrow}$ if $q_\uparrow = q_\downarrow = e$).
This means that semi-quantized quantum Hall states can also show non-Fermi liquid physics. The modified prefactor of the Chern-Simons term furthermore implies that the gapless composite quasiparticles should be interpreted as a liquid of anyons, rather than fermions, which further strengthens their non-Fermi liquid behaviour. To probe non-Fermi liquid physics in semi-quantized quantum Hall states, future experiments should address the scaling of the resistivity with temperature. Future theoretical work should in addition extend the analyses of Refs. [9–11] by including quasiparticles in the ↑-sector: if those are pinned to impurity sites, $\mathcal{L}_{CS}$ states that these quasiparticles generate non-trivial gauge-field disorder landscapes, which could provide a testbed for the ongoing characterization of many-body localization physics in two dimensions.

## 9.4 A note on thermal transport and the Wiedemann-Franz law

Besides electric transport, semi-quantized quantum Hall states also carry thermal currents. The available low-energy excitations that carry these thermal currents are the gapless composite quasiparticles of fractional charge $q_\downarrow^*$, and the fractional quantum Hall edge state. Due to the fractional nature of both of these, the Wiedemann-Franz law will in general be violated. How exactly the Wiedemann-Franz law is violated, however, depends on the measurement setup, e.g. which of the layers are connected electrically and/or thermally. In addition, the perfect interlayer drag in electric transport heavily relies on the fact that electrons cannot tunnel between layers, i.e. that there is no charge transfer between layers. While such a situation can experimentally be realized for charge transfer, heat is much more prone be transferred, or at least leak, across the intermediate layers in an experimental bilayer quantum Hall heterostructure. A detailed study of thermal and thermoelectric transport needs to take all of these effects into account, and is left for the future.

# 10   Summary and conclusions

In this work, we derived the low-energy theory of semi-quantized quantum Hall states in bilayer quantum Hall systems. We showed that both a heuristic continuum flux attachment picture and a more microscopic coupled-wire approach yield the same universal low-energy theory. This theory describes semi-quantized quantum Hall states by two sectors, a fully gapped one and a gapless one. The fully gapped sector protects topological-order like properties, such as the emergence of fractionally charged quasiparticles. Excitations in the gapless sector correspond to electrons in one layer glued to Laughlin-like quasiparticles in the other, and thus carry a net fractional charge. Because these composite gapless quasiparticles also couple to the emergent Chern-Simons gauge field deriving from the gapped sector, they can be thought of as forming an anyonic liquid. We analysed experimental signatures of semi-quantized quantum Hall and identified specific signatures in electric transport. We find Hall and Hall drag signals consistent with recent experiments [7]. Semi-quantized quantum Hall states are also promising platforms for novel non-Fermi liquid physics, and serve as versatile parent states for fully gapped phases with anyonic properties.

# Acknowledgments

We acknowledge financial support from the DFG via the Emmy Noether Programme ME 4844/1-1, SFB 1143 (project-id 247310070), and the Würzburg-Dresden Cluster of Excellence on Complexity and Topology in Quantum Matter - ct.qmat (EXC 2147, project-id 39085490). We are grateful for helpful discussions with A. Stern, A. G. Grushin, T. Neupert, K. Shtengel, and A. Rosch.

# Appendix: Braiding of quasiparticles in a gapped charge-density wave state

The gapless sector can be gapped by various mechanisms, one of which is the formation of a charge-density wave. In terms of the chiral quasiparticles $\Psi_{rj} = \exp\left\{-i\,\widetilde{\Theta}'_{rj\downarrow}\right\}$, a charge-density wave state is stabilized by interwire backscattering of quasiparticles, i.e. by $\Psi^\dagger_{Rj}\Psi_{Lj} + \text{h.c.} \sim \exp\left\{i\left(\widetilde{\Theta}'_{Rj\downarrow} - \widetilde{\Theta}'_{Lj\downarrow}\right)\right\} + \text{h.c.}$, giving rise to the sine-Gordon terms

$$g_{j,\text{CDW}}\cos\left(\widetilde{\Theta}'_{Rj\downarrow} - \widetilde{\Theta}'_{Lj\downarrow}\right) = g_{j,\text{CDW}}\cos\left(\Phi_{Rj\downarrow} - \Phi_{Lj\downarrow}\right). \tag{84}$$

If these sine-Gordon terms flow to strong coupling, the system becomes fully gapped (note that the argument of these sine-Gordon terms commute with themselves at different positions, and with the arguments of the sine-Gordon terms $g_{\uparrow,j+1/2}$). The full gap provides an inverse time-scale that protects braiding. Coupled-wire constructions describe braiding via strings of operators that hop quasiparticles around closed loops [2].

**Braiding of quasiparticles in $g_{k+1/2\uparrow}$ sine-Gordon terms.**   Quasiparticles in the $g_{k+1/2\uparrow}$ sine-Gordon terms can be moved from link $k-1/2$ to link $k+1/2$ by application of an exponential of

$$\Theta_{Rj\uparrow} - \Theta_{Lj\uparrow} = K^{-1}_{\uparrow\uparrow}\left(\widetilde{\Phi}_{Rj\uparrow} - \widetilde{\Phi}_{Lj\uparrow}\right) + K^{-1}_{\uparrow\downarrow}\left(\widetilde{\Phi}_{Rj\downarrow} - \widetilde{\Phi}_{Lj\downarrow}\right) = \Phi_{Rj\uparrow} - \Phi_{Lj\uparrow}. \tag{85}$$

It can be checked by calculating the commutator with the argument of the sine-Gordon term changes $\widetilde{\Phi}_{Rj\uparrow} - \widetilde{\Phi}_{Lj+1\uparrow}$ by $+2\pi$ and $\widetilde{\Phi}_{Rk-1\uparrow} - \widetilde{\Phi}_{Lj\uparrow}$ by $-2\pi$, while leaving the $\downarrow$-sector untouched.

Using the definitions of the different fields as well as $K_{\uparrow\uparrow}K_{\uparrow\uparrow}^{-1} + K_{\uparrow\downarrow}K_{\uparrow\downarrow}^{-1} = 1$, one can rewrite this operator as

$$\Theta_{Rj\uparrow} - \Theta_{Lj\uparrow} = \frac{1}{K_{\uparrow\uparrow}}\left(\widetilde{\Phi}_{Rj\uparrow} - \widetilde{\Phi}_{Lj\uparrow}\right) - \frac{K_{\uparrow\downarrow}}{K_{\uparrow\uparrow}}\left(\Phi_{Rj\downarrow} - \Phi_{Lj\downarrow}\right). \tag{86}$$

If we now look at a closed braiding path, the total phase we pick up is related to

$$\sum_j \left[\Theta_{Rj\uparrow} - \Theta_{Lj\uparrow}\right] = \sum_j \frac{1}{K_{\uparrow\uparrow}}\left(\widetilde{\Phi}_{Rj\uparrow} - \widetilde{\Phi}_{Lj\uparrow}\right) + \sum_j \left(-\frac{K_{\uparrow\downarrow}}{K_{\uparrow\uparrow}}\right)\left(\Phi_{Rj\downarrow} - \Phi_{Lj\downarrow}\right)$$
$$= \sum_j \frac{1}{K_{\uparrow\uparrow}}\left(\widetilde{\Phi}_{Rj\uparrow} - \widetilde{\Phi}_{Lj+1\uparrow}\right) + \sum_j \left(-\frac{K_{\uparrow\downarrow}}{K_{\uparrow\uparrow}}\right)\left(\Phi_{Rj\downarrow} - \Phi_{Lj\downarrow}\right), \tag{87}$$

where $\sum$ is discrete analog of line integral and means summation on closed loops. A quasiparticle corresponds to a $2\pi$-kink in the argument of one of the sine-Gordon terms. This tells us that braiding a quasiparticle in the $\uparrow$-sector around another one of these quasiparticles yields a phase

$$\varphi_{\uparrow\uparrow} = \frac{1}{K_{\uparrow\uparrow}}2\pi, \tag{88}$$

while braiding an $\uparrow$-quasiparticle around a $\downarrow$-quasiparticle yields a phase of

$$\varphi_{\uparrow\downarrow} = -\frac{K_{\uparrow\downarrow}}{K_{\uparrow\uparrow}}2\pi. \tag{89}$$

These braiding phases are consistent with the picture of the quasiparticles in the initially gapless $\downarrow$-sector as being composed of electrons in the bottom layer and Laughlin-like quasiparticles in the bottom layer, while the emergent gauge theory discussed in the main text shows that quasiparticles $\uparrow$-sector are similar to Laughlin quasiparticles.

**Braiding of quasiparticles in $g_{k,\text{CDW}}$ sine-Gordon terms.** Our discussion of Sec. 4.1 shows that quasiparticles in the $\downarrow$-sector are moved around by exponentials of

$$\widetilde{\Theta}'_{rj+1\downarrow} - \widetilde{\Theta}'_{r'j\downarrow} = \Phi_{Rj+1\downarrow} - \Phi_{r'k\downarrow} - x_4\left(\Phi_{Rj\uparrow} - \Phi_{Lj\uparrow}\right) - x_3\left(\Phi_{Rj+1\uparrow} - \Phi_{Lj+1\uparrow}\right). \tag{90}$$

One can check that this operator indeed moves a quasiparticle in the $\downarrow$-sector from wire $j$ to wire $j+1$ while leaving the $\uparrow$-sector untouched. The phase picked up by a quasiparticle in the $\downarrow$-sector is given by

$$\sum_j \left[\Phi_{rj+1\downarrow} - \Phi_{r'j\downarrow} - x_4\left(\Phi_{Rj\uparrow} - \Phi_{Lj\uparrow}\right) - x_3\left(\Phi_{Rj+1\uparrow} - \Phi_{Lj+1\uparrow}\right)\right]$$
$$= \sum_j \left[\Phi_{rj\downarrow} - \Phi_{r'k\downarrow} - K_{\uparrow\downarrow}\left(\Phi_{Rj\uparrow} - \Phi_{Lj\uparrow}\right)\right] = \sum_j \left[\Phi_{rj\downarrow} - \Phi_{r'k\downarrow} - K_{\uparrow\downarrow}\left(\Theta_{Rj\uparrow} - \Theta_{Lj\uparrow}\right)\right]$$
$$= \sum_j \left(\frac{\hat{r} - \hat{r}'}{2} + \frac{K_{\uparrow\downarrow}^2}{K_{\uparrow\uparrow}}\right)\left(\Phi_{Rj\downarrow} - \Phi_{Lj\downarrow}\right) + \sum_j \left(-\frac{K_{\uparrow\downarrow}}{K_{\uparrow\uparrow}}\right)\left(\widetilde{\Phi}_{Rj\uparrow} - \widetilde{\Phi}_{Lj+1\uparrow}\right). \tag{91}$$

This tells us that braiding a quasiparticle in the ↓-sector around another one of these quasi-particles yields a phase

$$\varphi_{\downarrow\downarrow} = \left( \frac{\hat{r} - \hat{r}'}{2} + \frac{K_{\uparrow\downarrow}^2}{K_{\uparrow\uparrow}} \right) 2\pi = \frac{K_{\uparrow\downarrow}^2}{K_{\uparrow\uparrow}} 2\pi \, \text{mod}(2\pi), \tag{92}$$

while braiding an ↓-quasiparticle around a ↑-quasiparticle yields a phase of

$$\varphi_{\downarrow\uparrow} = -\frac{K_{\uparrow\downarrow}}{K_{\uparrow\uparrow}} 2\pi. \tag{93}$$

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
