# Peer review of "Microscopic theory of fractional excitations in gapless quantum Hall states: semi-quantized quantum Hall states"

_SciPost Physics, doi:SciPost Phys. 8, 031 (2020)_

## Round 1 · Referee Report · Anonymous (Referee 1) · 2019-12-29

Strengths

  1. Nontrivial demonstration of the wire construction approach for a novel partially gapped strongly interacting state
  2. Theory immediately relevant for a recent experiment on bilayer graphene

Weaknesses

  1. It it unclear at this stage what are the advantages of the present wire construction approach over the composite fermion mean field theory, and what new does it tell us about this semi-quantized state from the approach.

Report

In this paper, Tuerker and Meng study strongly interacting fractional quantum Hall bi-layers, motivated by recent experiments on bilayer graphene where semi-quantized drag-Hall effects were detected and explained using a theory of composite fermions partially at compressible and incompressible filling factors.

The present paper applies the wire-construction approach to account for this partially gapped state. Using an impressive sequence of transformations and manipulations an effective theory is derived, from which the measured responses follow. This achievement is quite remarkable. I find the paper very well written and specifically the technical details are well presented allowing even nonexperts to grasp the essentials.

Given the immediate relevance to the recent experiment, I believe the technical results of this paper will be useful for further analysis of such exotic strongly correlated state. Thus I recommend its publication in SciPost Physics.

Yet I’d like to ask the authors to address the following questions.

While the reader motivated to learn about the derivation can find important details through the derivation steps, I believe that a broader introduction – written for the broader audience and explaining the important results of the present paper - would be highly beneficial.

The authors explain that one of the two sectors of the theory are gapless. In this case, how can one understand a quantized response? Why doesn't the gapless sector contributes an arbitrary correction that spoils quantization? (or equivalently, what does ‘semi-quantized’ response mean?)

Is it correct that the present approach is an extension of the composite fermion theory developed in Ref 54 using the complementary wire-construction approach? [This is what I get from the sentence: 'Our microscopic
derivation of the topological field theory governing these states goes beyond the composite
fermion picture of Ref. [54], but agrees with it where we overlap'.

To enhance the usefulness of this paper it would be important to clarify what could be the advantages of using the wire construction approach, apart from the statement that it provides a microscopic model.

Typos: Doesn't equation 2 miss an interaction between the $\downarrow$ layer densities?

operators that layer--> operators in that layer

quantum Hall state feature--> quantum Hall state features

---

## Round 1 · Referee Report · Emil Bergholtz (Referee 2) · 2020-1-9

Strengths

  1. Beautiful and sophisticated theory of direct relevance for intriguing recent experiments.
  2. Generally very pedagogical well written.

Weaknesses

  1. Somewhat sloppy referencing in the introduction.

Report

The manuscript "Microscopic theory of fractional excitations in gapless quantum Hall states: semi-quantized quantum Hall states” by Türker and Meng is impressive work superbly presented. I very much enjoyed reading the two very different ways of arriving at Eq. (12), which is the low-energy theory describing the semi-quantized quantum Hall states, and then the exploration of its experimental consequences in terms of transport. On its own this work would in my opinion easily meet the criteria from SciPost Physics based on the theoretical novelty despite the fact that several insights discussed were first reported in Ref. 54. Here, the direct correspondence made between the somewhat hand waving flux attachment picture and the more microscopic coupled wire construction is very illuminating, and so is the manner in which the various experimentally accessible probes discussed. That it is directly relevant for -- and seemingly consistent with -- the recent experiments of Ref. 54 on bilayer graphene systems, is a considerable strength of the present manuscript.

I also think it is highly likely that much work on semi-quantized quantum Hall states will follow and that this paper will generate much of that interest. Indeed, Türker and Meng outline several intriguing directions worth future investigation both experimentally and theoretically.

The only thing I found worth complaining about is the referencing in the introduction. It seem like the first few (three?) references are actually missing — there are simply question marks in the text and no sign of the references in the bibliography either. Moreover, I do not find it terribly helpful to write “see Refs. [3–53]” after providing a long list of different phenomena. It would be helpful for any reader if the references were instead split into shorter lists for each of the phenomena mentioned.

Requested changes

  1. Fix the references in the introduction.

---

## Round 4 · Author Response

List of changes
Referee 1
* * *
We thank the referee for their constructive feedback, and for the very positive assessment of our paper. In response to their helpful suggestions, we have made the following modifications in the introductory section 1.
- We added a new paragraph starting with "In a more general perspective" that puts our work into a bigger picture in order to address a broader readership.
- We explain why a partially gapless system can exhibit quantized transport. For this to happen, it is key that the gapless sector is electrically disconnected. As we now write, "We discuss that the persistence of quantized responses arises only if the layer associated with the gapless sector is electrically disconnected. As one might expect, driving a current through that layer would instead lead to a non-quantized response."
- It is correct that the present approach is an extension of the composite fermion theory developed in Ref 54 using the complementary wire-construction approach, which we now state explicitly.
- We also clarify the advantages of using the wire construction approach: "As the main advantage as compared to a continuum theory as the one in Sec. 2, our coupled-wire construction not only provides a microscopic model for semi-quantized quantum Hall states but also facilitates an exact translation of observables between the languages of the effective low-energy field theory and the original electronic operators".
To iterate, coupled-wire constructions are to be seen as an alternative technical implementation of flux attachment and the controlled derivation on the universal low-energy theory, not as an opposing physical picture. The general logic is that coupled-wire constructions describe anisotropic limits of topological states of matter. The states described by these constructions should be adiabatically connected to the isotropic states realized in quantum Hall systems for the approach to make sense. The anisotropic limit in then "buys" more explicit calculations, which in turn facilitates, for example, easier identification of the various operators in the low-energy theory in terms of the original degrees of freedom.
- We have corrected the typos found by the referee. Note that we do not have an interaction between the \down-electrons. Adding one would be straightforward and not affect the result of our heuristic argument, but would render the equations heavier. We now state this explicitly in the text.
Referee 2
* * *
We thank the referee for his assessment of our paper as being beautiful, sophisticated, and pedagogical at the same time, and stating that he feels like it will generate future interest in semi-quantized quantum Hall states. We apologize for the poor referencing in the earlier version of the paper and have corrected this now.

---

## Round 4 · List of Changes

Referee 1
* * *
We thank the referee for their constructive feedback, and for the very positive assessment of our paper. In response to their helpful suggestions, we have made the following modifications in the introductory section 1.
- We added a new paragraph starting with "In a more general perspective" that puts our work into a bigger picture in order to address a broader readership.
- We explain why a partially gapless system can exhibit quantized transport. For this to happen, it is key that the gapless sector is electrically disconnected. As we now write, "We discuss that the persistence of quantized responses arises only if the layer associated with the gapless sector is electrically disconnected. As one might expect, driving a current through that layer would instead lead to a non-quantized response."
- It is correct that the present approach is an extension of the composite fermion theory developed in Ref 54 using the complementary wire-construction approach, which we now state explicitly.
- We also clarify the advantages of using the wire construction approach: "As the main advantage as compared to a continuum theory as the one in Sec. 2, our coupled-wire construction not only provides a microscopic model for semi-quantized quantum Hall states but also facilitates an exact translation of observables between the languages of the effective low-energy field theory and the original electronic operators".
To iterate, coupled-wire constructions are to be seen as an alternative technical implementation of flux attachment and the controlled derivation on the universal low-energy theory, not as an opposing physical picture. The general logic is that coupled-wire constructions describe anisotropic limits of topological states of matter. The states described by these constructions should be adiabatically connected to the isotropic states realized in quantum Hall systems for the approach to make sense. The anisotropic limit in then "buys" more explicit calculations, which in turn facilitates, for example, easier identification of the various operators in the low-energy theory in terms of the original degrees of freedom.
- We have corrected the typos found by the referee. Note that we do not have an interaction between the \down-electrons. Adding one would be straightforward and not affect the result of our heuristic argument, but would render the equations heavier. We now state this explicitly in the text.
Referee 2
* * *
We thank the referee for his assessment of our paper as being beautiful, sophisticated, and pedagogical at the same time, and stating that he feels like it will generate future interest in semi-quantized quantum Hall states. We apologize for the poor referencing in the earlier version of the paper and have corrected this now.

---

## Editorial Decision

published